# Dynamic Mixture of Curriculum LoRA Experts for Continual Multimodal Instruction Tuning

**Chendi Ge** [1]  **Xin Wang** [1]  **Zeyang Zhang** [1]  **Hong Chen** [1]  **Jiapei Fan** [2]  **Longtao Huang** [2]  **Hui Xue** [2]  **Wenwu Zhu** [1]

## Abstract

Continual multimodal instruction tuning is crucial for adapting Multimodal Large Language Models (MLLMs) to evolving tasks. However, most existing methods adopt a fixed architecture, struggling with adapting to new tasks due to static model capacity. We propose to evolve the architecture under parameter budgets for dynamic task adaptation, which remains unexplored and imposes two challenges: 1) *task architecture conflict*, where different tasks require varying layer-wise adaptations, and 2) *modality imbalance*, where different tasks rely unevenly on modalities, leading to unbalanced updates. To address these challenges, we propose a novel **D**ynamic **M**ixture **o**f Curriculum **L**oRA **E**xperts (**D-MoLE**) method, which automatically evolves MLLM's architecture with controlled parameter budgets to continually adapt to new tasks while retaining previously learned knowledge. Specifically, we propose a dynamic layer-wise expert allocator, which automatically allocates LoRA experts across layers to resolve architecture conflicts, and routes instructions layer-wisely to facilitate knowledge sharing among experts. Then, we propose a gradient-based inter-modal continual curriculum, which adjusts the update ratio of each module in MLLM based on the difficulty of each modality within the task to alleviate the modality imbalance problem. Extensive experiments show that D-MoLE significantly outperforms state-of-the-art baselines, achieving a 15% average improvement over the best baseline. To the best of our knowledge, this is the first study of continual learning for MLLMs from an architectural perspective.

[1]Department of Computer Science and Technology, BNRist, Tsinghua University, Beijing, China [2]Alibaba Group, Hangzhou, China. Correspondence to: Xin Wang <xin_wang@tsinghua.edu.cn>, Wenwu Zhu <wwzhu@tsinghua.edu.cn>.

*Proceedings of the $42^{nd}$ International Conference on Machine Learning*, Vancouver, Canada. PMLR 267, 2025. Copyright 2025 by the author(s).

## 1. Introduction

Multimodal Large Language Models (MLLMs) (Dai et al., 2023; Liu et al., 2023; Bai et al., 2025; Zhu et al., 2025), which extend conventional LLMs through integration with modality-specific encoders (e.g., vision, audio), have demonstrated remarkable capabilities in processing heterogeneous multimodal data. However, in real-world scenarios, pretrained MLLMs will inevitably encounter new data as users' instructions and demands shift. To handle these new tasks, MLLMs need to be adapted, but this adaptation process can lead to catastrophic forgetting, where the model loses previously learned knowledge. As a result, Continual Multimodal Instruction Tuning (CMIT) has recently received considerable attention (Chen et al., 2024a), with the goal of efficiently adapting MLLMs to new tasks while preserving previously learned knowledge.

Despite recent progress, most existing methods adopt a *fixed architecture*, which limits model capacity and flexibility for both past and future tasks in continual learning. Common approaches include replay-based methods (Wang et al., 2024b; Song et al., 2023) and parameter regularization methods (Wang et al., 2023a; Xiang et al., 2023), both originally designed for smaller unimodal models with simpler input structures. However, MLLMs are structurally different: they consist of modality-specific encoders, projectors, and language models, forming a heterogeneous and layered architecture. The reliance on each component varies significantly across tasks, resulting in task-specific sensitivity across both layers and modalities. As a result, fixed-architecture MLLMs face increased difficulty in balancing knowledge retention and adaptation. These observations motivate us to move beyond static designs and explore dynamic architectural adaptation for CMIT.

To bridge this gap, in this paper, we propose to continually evolve the architecture under given parameter budgets in CMIT, aiming to adapt to new tasks with dynamic capacity while maintaining the previous knowledge, without significantly increasing computational resources. The problem, remaining unexplored in the literature, is highly non-trivial with the following key challenges:

*(1) Task architecture conflict.* In CMIT, different tasks ex-

hibit varying information patterns, leading to varying sensitivities across transformer layers in MLLM. This variation makes it challenging to determine where to apply architecture evolution, as certain layers may be more critical for some tasks than others.

*(2) Modality imbalance.* In CMIT, the reliance on different modalities varies by task, which may cause one modality to dominate the learning process. This imbalance poses challenges in achieving balanced updates across modalities, often resulting in under-optimized training for the modules of different modalities in the MLLM.

To address these challenges, we propose a novel **D**ynamic **M**ixture **o**f Curriculum **L**oRA **E**xperts (**D-MoLE**) method for CMIT, which is able to automatically evolve MLLM's architecture within controlled budgets to continually adapt to new tasks without forgetting previously learned knowledge. Specifically, we propose a dynamic layer-wise expert allocator, which automatically allocates LoRA experts across layers to resolve architecture conflicts. This module, based on scores generated by training-free zero-cost proxies, optimizes resource allocation by applying model evolution to the most critical layers for each task, and routes instructions layer-wisely to facilitate knowledge sharing among experts. Then, we propose a gradient-based inter-modal continual curriculum, which dynamically adjusts the update ratio between the language model and modality encoder during the model's architecture evolution based on each task's difficulty for each modality, mitigating the modality imbalance issue. By dynamically allocating LoRA experts and using curriculum to guide inter-modal optimization, D-MoLE provides a scalable and efficient solution for continual learning of MLLMs. Extensive experiments demonstrate that D-MoLE significantly outperforms state-of-the-art baselines, achieving a 15% average performance improvement over the best baseline. To the best of our knowledge, this is the first study of continual learning for MLLMs from an architectural perspective.

The contributions of our work are summarized as follows:

- We introduce the Dynamic Mixture of Curriculum LoRA Experts (D-MoLE), the first method to study continual learning for MLLMs from an architectural perspective. D-MoLE evolves the MLLM's architecture within controlled budgets, enabling it to continually adapt to new tasks while retaining previously learned knowledge.

- We observe the phenomena of *task architecture conflict* and *modality imbalance* in CMIT. For the *task architecture conflict*, we provide a theoretical analysis showing that different layers exhibit varying sensitivities to different tasks, making uniform resource allocation inefficient.

- We propose a *dynamic layer-wise expert allocator* that automatically assigns experts layer-wise across tasks, ad-

dressing the *task architecture conflict* under constrained parameter budgets, and a *gradient-based inter-modal continual curriculum* that dynamically adjusts the updating between multimodal modules, effectively mitigating the *modality imbalance*.

- Extensive experiments show that D-MoLE significantly outperforms state-of-the-art methods in CMIT, achieving a 15% average improvement over the best baseline in task adaptation and knowledge retention.

## 2. Problem Formulation

### 2.1. Continual Multimodal Instruction Tuning

Continual Multimodal Instruction Tuning (CMIT) refers to adapting MLLMs to sequential tasks while retaining both general and task-specific knowledge. Let $\{\mathcal{T}_1, \mathcal{T}_2, \ldots, \mathcal{T}_N\}$ denote the tasks, and $\{\mathcal{D}_1, \mathcal{D}_2, \ldots, \mathcal{D}_N\}$ their corresponding instruction data. At each time step $i$, the model receives a new dataset $\mathcal{D}_i$ and integrates new knowledge without forgetting prior tasks. Previously encountered data $\{\mathcal{D}_k\}_{k=1}^{i-1}$ is generally inaccessible, except in replay-based methods that store selected samples in a memory buffer.

In CMIT, each dataset $\mathcal{D}_i = \{(\mathbf{t}_j^i, \mathbf{v}_j^i, \mathbf{o}_j^i)\}_{j=1}^{N_i}$ consists of textual inputs $\mathbf{t}$, visual inputs $\mathbf{v}$, and outputs $\mathbf{o}$. The MLLM processes these multimodal instructions to generate outputs while avoiding catastrophic forgetting. The key challenge is balancing generalization across past tasks with fine-tuning for new tasks while managing the complexities of multimodal data.

### 2.2. CMIT with Architecture Evolution

In this paper, we explore an architecture evolution-based approach for MLLMs' continual learning. Given the limited computational resources in real-world settings, we cannot simply introduce a large number of new parameters for each task. Therefore, building on the CMIT framework outlined in Section 2.1, we define a parameter budget $B_{\text{total}}$ for each task. Our problem formulation focuses on how to dynamically allocate these parameters to the most important layers within the MLLM under this parameter budget, ensuring efficient adaptation to new tasks.

## 3. Preliminary Study

Based on the problem formulation introduced in Section 2.2, which focuses on optimally allocating newly introduced parameters in the CMIT scenario, there are two key challenges: *task architecture conflict*, arising from the fact that the optimal positions for parameter allocation differ across tasks, and *modality imbalance*, where different modalities have varying levels of dominance across tasks. In this section, we present our key observations regarding these challenges,

offering insights into their impact on CMIT and how they guide the design of our proposed method.

### 3.1. Task Architecture Conflict in CMIT

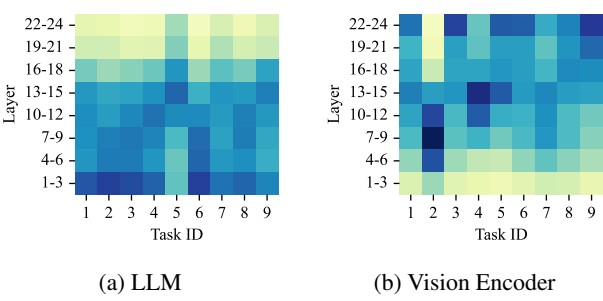

(a) LLM        (b) Vision Encoder

Figure 1: Sensitivity of different transformer layers across various tasks for LLM and vision encoder in MLLM. Darker colors indicate higher sensitivity. The order of the task is the same as in Table 2.

In existing MLLMs, both the LLM and modality encoders consist of multiple stacked transformer layers, with each layer contributing differently to tasks due to varying feature abstraction requirements. Lower layers typically process basic features, while higher layers handle abstract representations (Zhang et al., 2024d; Gao et al., 2024). As a result, different tasks tend to rely on different subsets of layers for effective representation, especially when tasks differ in modality focus or semantic complexity.

We quantify this *task architecture conflict* using the gradient norm metric (Abdelfattah et al., 2021), which measures the sensitivity of each transformer layer to different tasks. As shown in Figure 1, the sensitivity distribution varies across tasks and layers. In some cases, the lower layers in the vision encoder are more critical, while in others, higher LLM layers contribute more. This variability suggests that uniformly adding LoRA modules across layers does not effectively capture task-specific adaptation requirements.

Such uniform allocation leads to redundancy in less-relevant layers and under-adaptation in critical ones, resulting in inefficient use of parameters and degraded performance. Instead, parameter allocation should prioritize layers that contribute most to current task learning. We further support this observation with a formal theoretical analysis in Appendix K, showing that tasks induce nontrivial gradient discrepancies across layers.

**Theorem 3.1.** *Consider an MLLM with $L$ transformer layers trained sequentially on two distinct tasks $\mathcal{T}_A$ and $\mathcal{T}_B$. Under the assumption of task heterogeneity and non-collinear gradients, there exists at least one layer $l^* \in \{1, \ldots, L\}$ where the expected gradient norms differ:*

$$\left\| \mathbb{E}_{\mathcal{T}_A} \left[ \nabla_{\mathbf{W}_{l^*}} \mathcal{L} \right] \right\|_2 \neq \left\| \mathbb{E}_{\mathcal{T}_B} \left[ \nabla_{\mathbf{W}_{l^*}} \mathcal{L} \right] \right\|_2, \quad (1)$$

*where $\nabla_{\mathbf{W}_l} \mathcal{L}$ denotes the gradient of loss $\mathcal{L}$ with respect to parameters $\mathbf{W}_l$ at layer $l$.*

### 3.2. Modality Imbalance in CMIT

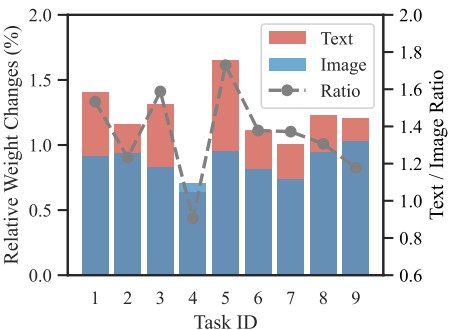

Figure 2: Relative dynamics for each task after LoRA fine-tuning. The order of the tasks is the same as in Table 2.

Modality imbalance refers to a phenomenon in multimodal learning where dominant modalities suppress weaker ones, leading to under-optimization of certain modules during training. This issue has been identified in smaller fusion-based models using probing or freezing techniques (Zhou et al., 2023; Du et al., 2021; Peng et al., 2022), and remains a concern in larger MLLMs (Wu et al., 2024a). However, due to the tightly coupled structure in MLLMs, directly probing individual modalities becomes nontrivial.

To characterize modality imbalance in CMIT, we track the relative weight change (i.e., gradient update magnitude) for the LLM and vision encoder across tasks. As shown in Figure 2, the update dynamics vary: in some tasks the LLM dominates, while in others the visual encoder exhibits stronger updates. This suggests that modality importance is task-dependent, and static allocation of adaptation budget across modalities is insufficient.

Consequently, a dynamic mechanism is needed to adjust update ratios based on task-specific modality dependence. This motivates our use of gradient-based inter-modal curriculum, which allocates resources adaptively based on modality sensitivity estimated from training-free proxies.

## 4. Methodology

The overall framework of D-MoLE is shown in Figure 3. D-MoLE comprises two key components: the dynamic layer-wise expert allocator and the gradient-based inter-modal curriculum. This section introduces the details of each module, followed by the overall pipeline of our method.

### 4.1. Dynamic Layer-Wise Expert Allocator

To address the task architecture conflict issue in CMIT, introduced in Section 3.1, we propose the *Dynamic Layer-Wise*

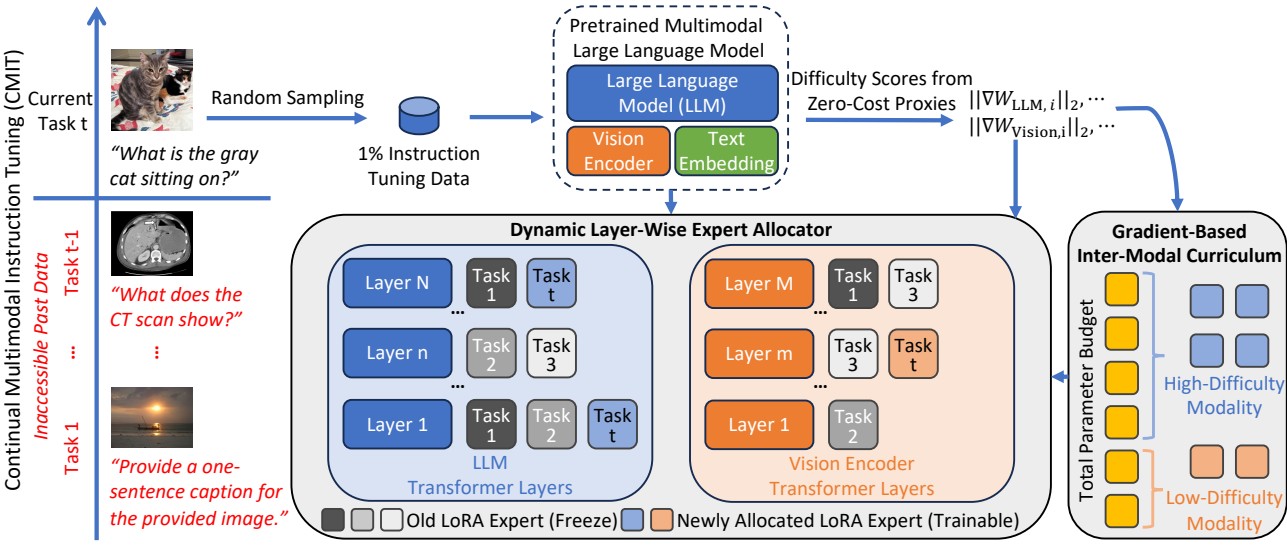

Figure 3: Overall framework of D-MoLE. During training on task $t$, all data from previous tasks (1 to $t-1$) remains inaccessible. D-MoLE first samples a small subset of multimodal instruction tuning data (e.g., 1%) from the current task's dataset. These samples are used to compute difficulty scores for each transformer layer in both the LLM and vision encoder using training-free zero-cost proxies. The difficulty scores then guide the gradient-based inter-modal curriculum and determine the optimal layer-wise expert allocation architecture for the current task. The parameters of the pretrained MLLM and the LoRA experts from previous tasks remain frozen, while only the newly allocated LoRA experts are updated.

*Expert Allocator* module. This module leverages training-free zero-cost proxies to identify which transformer layers are most critical for the current task, thereby determining where to introduce new parameters.

**Dynamic LoRA Expert**   In continual learning, when a new task $\mathcal{T}_t$ arrives, we need to update the MLLM model parameters to adapt to the new task. To reduce the number of new parameters introduced for each task, we use LoRA (Low-Rank Adaptation) (Hu et al., 2022), one of the most widely adopted methods for parameter-efficient fine-tuning (PEFT) (Ding et al., 2023). However, simply applying LoRA in the continual learning setting faces the task architecture conflict. Therefore, we propose *Dynamic Mixture of Curriculum LoRA Experts* (D-MoLE) for CMIT. Specifically, to adapt to the new task $\mathcal{T}_t$, the output of layer $l$ in the proposed D-MoLE is formulated as

$$f_l^t(\mathbf{x}) = \mathbf{W}_l^0 \mathbf{x} + \sum_{k=1}^{t-1} I_l^k \cdot g_l^k(\mathbf{x}) \cdot \Delta \mathbf{W}_l^k \mathbf{x} + I_l^t \cdot \Delta \mathbf{W}_l^t \mathbf{x},$$
$$\Delta \mathbf{W}_l^i = \mathbf{B}_l^i \mathbf{A}_l^i, \quad \text{for } i \in \{1, \ldots, t\},$$
$$(2)$$

where the pretrained MLLM weights at layer $l$ are denoted by $\mathbf{W}_l^0$. The binary indicator $I_l^k$ determines whether a LoRA expert is allocated to layer $l$ for task $k$, and the gating function $g_l^k(\mathbf{x})$ controls the activation of the $k$-th expert based on the input $\mathbf{x}$. The low-rank update for task $i$ is given by $\Delta \mathbf{W}_l^i = \mathbf{B}_l^i \mathbf{A}_l^i$, where $\mathbf{B}_l^i$ and $\mathbf{A}_l^i$ are trainable

matrices in LoRA.

The first term represents the output from the pretrained MLLM weights, while the third term provides the update for adapting to the current task $\mathcal{T}_t$ using the newly allocated LoRA expert. The second term aggregates contributions from LoRA experts trained on previous tasks. Inspired by Mixture of LoRA Experts (MoLE) (Wu et al., 2024c), we use this term to enable knowledge transfer from prior tasks.

This formulation allows the model to reuse prior knowledge while adapting to new tasks. The gating function helps select relevant experts, and layer-wise allocation provides additional flexibility.

Next, we discuss how we obtain the layer-wise indicator $I_l^t$ with the zero-cost proxies and how we obtain the gating function $g_l^k(\cdot)$ with the Autoencoder Router.

**Zero-Cost Proxies**   Prior to the training of the new task, we randomly sample a small subset of the training data for task $\mathcal{T}_t$, denoted as $\mathcal{D}_{\text{sub},t}$ (e.g., 1% of the full dataset), and use it to compute the importance score via zero-cost proxies. Specifically, we utilize the gradient norm as a zero-cost proxy, applied to the pretrained model. The entire model remains unfrozen, and we perform one forward and one backward pass on this small subset $\mathcal{D}_{\text{sub},t}$ to calculate the $L_2$-norm of the gradients for each layer, without updating any parameters.

The gradient norm for a given layer $l$ during task $\mathcal{T}_t$ is

computed as

$$\|\nabla \mathbf{W}_l^t\|_2 = \left\| \frac{\partial \mathcal{L}(\mathcal{D}_{\text{sub},t})}{\partial \mathbf{W}_l^0} \right\|_2, \quad (3)$$

where $\mathcal{L}(\mathcal{D}_{\text{sub},t})$ is the loss function computed on $\mathcal{D}_{\text{sub},t}$, $\mathbf{W}_l^0$ is the pretrained weight of layer $l$, and $\frac{\partial \mathcal{L}(\mathcal{D}_{\text{sub},t})}{\partial \mathbf{W}_l^0}$ is the gradient of the loss with respect to that weight computed on the sub dataset $\mathcal{D}_{\text{sub},t}$.

**Layer-Wise Expert Allocation Indicator**   Based on the computed gradient norms for task $\mathcal{T}_t$, we rank the transformer layers according to their sensitivity (i.e., the magnitude of the gradient) for both the LLM and the vision encoder. Let $\mathcal{R}_M^t$ represent the ranked list of layers for module M $\in$ {LLM, Vision} during task $\mathcal{T}_t$.

We define a binary indicator variable $I_l^t$ for each layer $l$, defined as

$$I_l^t = \begin{cases} 1, & \text{if a LoRA expert is allocated to layer } l, \\ 0, & \text{otherwise.} \end{cases} \quad (4)$$

The allocation of LoRA experts is subject to budget constraints $B_M^t$ for module M in task $\mathcal{T}_t$. LoRA experts are assigned to the top-ranked layers within these budgets. Specifically, a LoRA expert is allocated to a layer $l$ during task $\mathcal{T}_t$ based on the following criteria, defined by

$$I_{M,l}^t = \mathbb{I}\left(\text{rank}(l, \mathcal{T}_t) \leq B_M^t \text{ for } l \in \mathcal{R}_M^t\right), \quad (5)$$

where $\mathbb{I}$ is the characteristic function, which outputs a 1 if the condition is met and a 0 otherwise, $\text{rank}(l, \mathcal{T}_t)$ denotes the rank of layer $l$ in the ordered list of layers based on their gradient norms $\|\nabla \mathbf{W}_l^t\|$ for task $\mathcal{T}_t$.

This layer-specific allocation strategy ensures that LoRA modules are placed where they are most needed, based on task-specific gradient signals. Each expert can be viewed as a modular unit specialized for one task (Wang et al., 2025; Pan et al., 2025), and their placement is automatically determined without manual tuning, aligning with the automated model design paradigm (Zhang et al., 2021; Ge et al., 2025).

**Autoencoder Router and Gating Function**   To support modular expert routing across tasks, we introduce a set of task-specific autoencoders. Collectively, they serve two complementary purposes: during the training phase, they identify the most relevant expert from previous tasks to enable knowledge transfer; during the evaluation phase, they determine expert activation without access to task identity and detect whether the input belongs to an unseen task via reconstruction thresholds, in which case the model falls back to the pretrained backbone.

Each task is associated with a lightweight 2-layer MLP autoencoder, consisting of an encoder and decoder with shared input/output dimensions. To train the autoencoder, we extract representative features from each multimodal instruction sequence: image features $\mathbf{v}$ from the vision encoder and text features $\mathbf{w}$ from the LLM. Max-pooling is applied to obtain fixed-length vectors:

$$\mathbf{v}_{\text{pooled}} = \max_{i=1,...,N} \mathbf{v}_i, \quad \mathbf{w}_{\text{pooled}} = \max_{i=1,...,M} \mathbf{w}_i, \quad (6)$$

which are concatenated as

$$\mathbf{z} = \text{concat}(\mathbf{v}_{\text{pooled}}, \mathbf{w}_{\text{pooled}}), \quad (7)$$

and used as input to the autoencoder. The reconstructed output is

$$\hat{\mathbf{z}}^t = \text{Autoencoder}^t(\mathbf{z}), \quad (8)$$

which is trained to minimize the mean squared reconstruction loss:

$$\mathcal{L}_{\text{rec}}^t(\mathbf{z}) = \mathcal{L}_{\text{MSE}}(\mathbf{z}, \hat{\mathbf{z}}^t). \quad (9)$$

Lower reconstruction error implies stronger similarity to task $t$, allowing the autoencoder to capture its specific data distribution.

During training, since the task identity is known, we pass data from $\mathcal{D}_t$ through all previous autoencoders $\{\text{Autoencoder}^k\}_{k=1}^{t-1}$ and identify the prior task with the lowest reconstruction error. The corresponding expert is activated along with the new expert for $\mathcal{T}_t$ to promote transfer.

During evaluation, task identity is unknown. We compute reconstruction losses across all autoencoders and identify relevant tasks using their thresholds:

$$\mathcal{R} = \{t \mid \mathcal{L}_{\text{rec}}^t(\mathbf{z}) \leq \tau_t\}. \quad (10)$$

These candidate tasks are ranked by reconstruction error:

$$\text{Rank}(t) = \text{Rank}(\mathcal{L}_{\text{rec}}^t(\mathbf{z})), \quad t \in \mathcal{R}, \quad (11)$$

and the top-ranked ones are selected for expert activation. The gating function then determines activation at each layer:

$$g_l^k(\mathbf{x}) = \begin{cases} 1, & \text{if } t_k \in \text{TopK}(\text{Rank}(t), 2), \\ 0, & \text{otherwise.} \end{cases} \quad (12)$$

This enables input-adaptive expert selection and falls back to the pretrained backbone when the input does not match any known task.

### 4.2. Gradient-Based Inter-Modal Continual Curriculum

To mitigate the modality imbalance issue in CMIT, introduced in Section 3.2, we present a *Gradient-Based Inter-Modal Continual Curriculum* that automatically adjusts the update ratio between the modalities when a new task arrives.

Table 1: A summary of dataset statistics.

| Task Type | Dataset Name | Task Description | Train Size | Test Size |
|---|---|---|---|---|
| **Visual Question Answering (VQA)** | VizWiz-VQA (Gurari et al., 2018) | Natural Image VQA | 20,523 | 4,319 |
| | IconQA (Lu et al., 2021) | Diagram Understanding VQA | 18,946 | 6,315 |
| | OCR-VQA (Mishra et al., 2019) | Text Understanding VQA | 166,043 | 100,037 |
| | KVQA (Shah et al., 2019) | Knowledge-based VQA | 146,408 | 24,294 |
| | PMC-VQA (Zhang et al., 2024a) | Medical VQA | 176,948 | 2,000 |
| **Image Captioning** | VizWiz-Caption (Gurari et al., 2020) | Natural Image Captioning | 117,155 | 7,750 |
| | TextCaps (Sidorov et al., 2020) | Text Recognition Captioning | 109,765 | 3,166 |
| | Flickr30k (Young et al., 2014) | General Image Captioning | 145,000 | 1,000 |
| **Visual Grounding** | SK-VG (Chen et al., 2023b) | Scene Knowledge Visual Grounding | 23,404 | 6,598 |

**Task Difficulty Assessment**    Inspired by curriculum learning (Wang et al., 2021), we adjust the update ratio between the LLM and the vision encoder by measuring the difficulty of each component in relation to the current task. Specifically, we use the gradient norm as the difficulty score, where M represents a module in the MLLM, and $M \in \{\text{LLM}, \text{Vision}\}$. The difficulty score of each module M for the current task $\mathcal{T}_t$ can be formulated as

$$\text{Score}_M^t = \left\| \frac{\partial \mathcal{L}(\mathcal{D}_{\text{sub},t})}{\partial \mathbf{W}_M^0} \right\|_2, \tag{13}$$

where $\text{Score}_M^t$ represents the difficulty score of module M, based on the gradient magnitude of its pretrained weights $\mathbf{W}_M^0$ with respect to the current task $\mathcal{T}_t$.

A larger score indicates that a module is more sensitive to the current task and requires more updates. This dynamic adjustment ensures that both the LLM and the vision encoder receive updates based on the task's requirements, balancing the adaptation effort based on their scores.

**Modality Update Ratio**    Given a total parameter budget $B_{\text{total}}$ for each task, we allocate LoRA experts to each module M based on its difficulty score. This allocation ensures that the parameter budget is dynamically distributed according to each module's sensitivity to the new task. The ratios directly determine the number of LoRA experts assigned to each module, which are further refined at the layer level.

The ratio of LoRA experts assigned to each module under the current task $\mathcal{T}_t$ is computed as

$$r_M^t = \frac{\text{Score}_M^t}{\text{Score}_{\text{LLM}}^t + \text{Score}_{\text{Vision}}^t}. \tag{14}$$

**Modality Parameter Budget**    Based on these ratios, the number of new LoRA experts allocated to module M is

$$B_M^t = r_M^t \cdot B_{\text{total}}. \tag{15}$$

The new parameter budgets for the LLM, $B_{\text{LLM}}^t$, and the vision encoder, $B_{\text{Vision}}^t$, are then applied to the dynamic

layer-wise expert allocator described in Section 4.1, which further refines the allocation strategy at the layer level, determining how the experts are distributed across the layers.

This inter-modal curriculum adapts both the LLM and the vision encoder to the specific demands of task $\mathcal{T}_t$, prioritizing the modality with greater task difficulty. By focusing adaptation on the more sensitive module, this approach ensures continual balancing between modalities.

### 4.3. Overall Pipeline of D-MoLE

When a new task $\mathcal{T}_t$ arrives, D-MoLE follows the pipeline described below. The full algorithm is in Appendix C.

**Training Phase**    We first sample a small subset $\mathcal{D}_{\text{sub},t}$ to compute zero-cost scores (Eq. (3)), which determine the task-specific modality difficulty and allocate the parameter budget $B_M^t$ between the LLM and modality encoder (Eq. (14) and Eq. (15)). The dynamic layer-wise expert allocator identifies the critical layers in both the LLM and modality encoder (Eq. (11) and Eq. (4)), after which LoRA experts are assigned to the top layers. Sequence features of $\mathcal{D}_{\text{sub},t}$ are extracted (Eq. (7)) and used to train an autoencoder that helps select previous experts for knowledge transfer. During task adaptation, the pretrained MLLM weights $\mathbf{W}^0$ and previously allocated LoRA experts remain frozen, with only newly allocated experts trained. Both the task-specific expert and the most relevant expert from prior tasks, determined by reconstruction error rankings $\mathcal{L}_{\text{rec}}^t(\mathbf{z})$, are used for computation.

**Evaluation Phase**    After training, the model is evaluated across tasks $\mathcal{T}_1$ to $\mathcal{T}_N$. If the task identifier is unknown, sequence embeddings $\mathbf{z}$ are passed through task-specific autoencoders to calculate reconstruction losses $\mathcal{L}_{\text{rec}}^t(\mathbf{z})$. The top-2 most relevant tasks are selected using the ranking of reconstruction errors $\text{Rank}(t)$ to process each data sample. If a reconstruction error exceeds the threshold $\tau_i$ for any autoencoder $i$, the sample is considered to belong to an unknown task and is processed by the pretrained MLLM.

Table 2: AVG, Last, and BWT scores (%) of various methods on the CMIT benchmark. The best results are highlighted in **bold**, and the second best results are underlined.

| Method | VizWiz-Cap | SK-VG | TextCaps | IconQA | OCR-VQA | Flickr30k | VizWiz-VQA | KVQA | PMC-VQA | *Average* |
|---|---|---|---|---|---|---|---|---|---|---|
| Zero-shot | 50.54 | 19.61 | 61.25 | 50.49 | 44.17 | 66.15 | 47.38 | 42.63 | 37.40 | 46.62 |
| Finetune | 151.36 | 65.66 | 138.69 | 96.75 | 65.00 | 87.80 | 68.92 | 49.45 | 49.90 | 85.95 |
| Joint-learning | 151.48 | 60.50 | 137.14 | 87.90 | 63.30 | 86.08 | 69.11 | 48.60 | 44.90 | 83.22 |
| **AVG** | | | | | | | | | | |
| Seq-FT | 80.86 | 25.42 | 92.33 | 69.06 | 49.62 | 67.37 | 47.01 | 40.74 | 35.59 | 56.45 |
| LwF-LoRA | 67.86 | 16.27 | 86.29 | 68.73 | 46.54 | 67.10 | 52.93 | 41.85 | 38.47 | 54.00 |
| EWC-LoRA | 74.82 | 30.45 | 94.95 | 75.30 | 48.46 | 72.56 | 51.56 | 40.75 | 38.28 | 58.57 |
| Dense MoLE | 71.28 | 27.91 | 92.24 | 68.46 | 47.40 | 73.86 | 52.62 | 41.58 | 38.62 | 57.11 |
| Sparse MoLE | 69.56 | 18.72 | 90.98 | 69.18 | 49.62 | 68.98 | 44.92 | **41.95** | 37.46 | 54.60 |
| MoLA | 75.06 | 22.47 | 90.04 | 70.44 | 47.34 | 74.96 | 48.75 | 40.31 | 37.84 | 56.36 |
| O-LoRA | 99.07 | 35.25 | 81.24 | 69.26 | 46.67 | 71.76 | 48.17 | 40.56 | 37.16 | 58.79 |
| **D-MoLE** | **148.19** | **57.29** | **115.92** | **78.62** | **53.64** | **77.73** | **53.37** | 40.87 | **39.23** | **73.87** |
| **Last** | | | | | | | | | | |
| Seq-FT | 52.70 | 21.76 | 72.22 | 62.26 | 50.15 | 77.24 | 62.03 | **49.95** | 48.95 | 55.25 |
| LwF-LoRA | 56.72 | 3.70 | 68.31 | 63.78 | 44.95 | 77.74 | 65.20 | 48.50 | 47.75 | 52.96 |
| EWC-LoRA | 59.53 | 22.50 | 72.27 | 82.60 | 48.20 | **80.48** | 68.18 | 49.10 | 45.95 | 58.76 |
| Dense MoLE | 50.25 | 21.13 | 75.20 | 58.53 | 48.75 | 77.55 | 65.62 | 49.55 | 48.40 | 55.00 |
| Sparse MoLE | 59.93 | 19.12 | 66.82 | 55.57 | 45.25 | 61.62 | 47.85 | 47.40 | 46.55 | 50.01 |
| MoLA | 61.96 | 18.76 | 81.96 | 65.45 | 47.20 | 78.71 | 63.08 | 48.85 | **49.55** | 57.28 |
| O-LoRA | 92.38 | 31.30 | 80.26 | 71.69 | 47.85 | 78.46 | 63.75 | 47.90 | 44.80 | 62.04 |
| **D-MoLE** | **148.77** | **61.60** | **131.41** | **93.05** | **62.85** | 78.32 | **69.16** | 46.05 | 48.40 | **82.18** |
| **BWT** | | | | | | | | | | |
| Seq-FT | -79.89 | -42.95 | -49.41 | -23.33 | -10.84 | -20.65 | -8.45 | **-0.10** | - | -29.45 |
| LwF-LoRA | -94.52 | -39.59 | -50.62 | -16.47 | -12.10 | -13.73 | -4.05 | -0.45 | - | -28.94 |
| EWC-LoRA | -86.68 | -34.27 | -43.90 | -8.77 | -12.12 | -7.76 | **-2.64** | -0.25 | - | -24.55 |
| Dense MoLE | -91.11 | -40.76 | -49.39 | -25.42 | -12.48 | -11.26 | -6.54 | -1.35 | - | -29.79 |
| Sparse MoLE | -95.91 | -52.64 | -53.43 | -22.43 | -6.49 | -30.43 | -22.53 | -2.50 | - | -35.79 |
| MoLA | -83.51 | -45.38 | -31.76 | -18.54 | -12.09 | -12.22 | -7.19 | -1.20 | - | -26.49 |
| O-LoRA | -59.41 | -17.14 | -56.26 | -11.87 | -10.68 | -11.44 | -2.91 | -0.75 | - | -21.31 |
| **D-MoLE** | **-0.06** | **0.01** | **-2.64** | **-1.55** | **-2.23** | **-0.25** | -3.48 | -1.70 | - | **-1.49** |

## 5. Experiment

In this section, we evaluate our proposed D-MoLE on the CMIT benchmark, providing a detailed introduction of the baselines, datasets, evaluation metrics, and experimental results. To demonstrate the effectiveness of each component, we also conduct ablation studies. Due to space constraints, detailed implementation information for both our method and the baselines can be found in Appendix A.

### 5.1. Experimental Setups

**Datasets**   To validate our method, we construct a comprehensive CMIT benchmark consisting of datasets from three primary task categories: Visual Question Answering (VQA),

Image Captioning, and Visual Grounding, as summarized in Table 1. The detailed construction process of the benchmark is provided in Appendix E.

**Baselines**   We compare our method with several baselines, all implemented using LoRA with a consistent constraint on the total number of trainable parameters for fair comparison. **Seq-FT** serves as the vanilla baseline, applying sequential fine-tuning to the model. **LwF-LoRA** (Li & Hoiem, 2017) using knowledge distillation to retain performance on prior tasks. **EWC-LoRA** (Xiang et al., 2023) is a LoRA-based implementation of EWC (Kirkpatrick et al., 2017), which penalizes changes to critical parameters to mitigate forgetting. **Dense MoLE** (Chen et al., 2024a) employs dense expert routing to address forgetting, while **Sparse MoLE** (Dou

et al., 2024) uses sparse expert selection. **MoLA** (Gao et al., 2024) builds on Sparse MoLE by allocating more experts to higher layers, and **O-LoRA** (Wang et al., 2023a) introduces orthogonal constraints to reduce task interference.

**Evaluation Metrics** For each task in CMIT, we use task-specific metrics: top-1 accuracy for VQA, CIDEr score (Vedantam et al., 2015) for Image Captioning, and Intersection over Union (IoU) for Visual Grounding, where a prediction is considered correct if the IoU exceeds 0.5.

To evaluate continual learning performance, we adopt the *AVG*, *Last*, and *Backward Transfer (BWT)* metrics. These assess both overall and task-wise knowledge retention, where $A_{t,i}$ denotes the evaluation score (e.g., accuracy, CIDEr, or IoU) on task $i$ after training on task $t$:

- **AVG**: The average score on task $i$ across all stages:

$$\text{AVG}_i = \frac{1}{N} \sum_{t=1}^{N} A_{t,i}. \tag{16}$$

- **Last**: The score on task $i$ after training the last task:

$$\text{Last}_i = A_{N,i}. \tag{17}$$

- **BWT**: Measures forgetting by comparing post-training performance to the initial post-task score:

$$\text{BWT}_i = \frac{1}{N-i} \sum_{t=i+1}^{N} (A_{t,i} - A_{i,i}). \tag{18}$$

Since we start from a pretrained MLLM, preserving general knowledge and zero-shot capability is crucial. We therefore evaluate all tasks after each training step, regardless of whether they have been encountered, following the evaluation setting in (Yu et al., 2024).

## 5.2. Experimental Results

**Main Results** Table 2 presents the AVG, Last, and BWT scores of different methods on our CMIT benchmark. Zero-shot refers to the pretrained model's performance on each task without fine-tuning. Finetune involves applying LoRA fine-tuning on each task individually from the pretrained model, outside a CL setting. Joint-learning represents the upper bound for CL, where instructions from all tasks are combined and trained together, offering the best-case scenario without task-ordering constraints. The key observations are as follows:

- D-MoLE outperforms most state-of-the-art approaches across the three metrics on most datasets. Compared

Table 3: Evaluation results on general MLLM benchmarks.

| Method | MME-Sum | MMMU-Val | POPE-Sum |
|---|---|---|---|
| Zero-Shot | 1876.8 | 35.4 | 87.3 |
| Seq-FT | 1513 | 30.7 | 85.8 |
| O-LoRA | 1646 | 32.3 | 87.6 |
| **D-MoLE** | **1754.6** | **32.7** | **88.1** |

to the second-best method, it achieves average improvements of 15.08%, 20.14%, and 19.82% on Average, Last, and BWT respectively. This demonstrates the effectiveness of our approach in mitigating forgetting over the long-term CMIT process.

- Compared to other MoLE-based methods, MoLA, which selectively allocates more LoRA experts to higher layers, outperforms Sparse MoLE and Dense MoLE. This result suggests that selective allocation of LoRA experts can help in model learning.

- Traditional CL methods like LwF-LoRA and EWC-LoRA, even when implemented with the PEFT approach (LoRA), struggle significantly in the CMIT setting. They introduce substantial computational overhead, such as distillation loss, making them slower to adapt to new knowledge. In contrast, O-LoRA, which applies orthogonal constraints to LoRA experts for each task, shows better performance. However, since it continues to add LoRA adapters without leveraging task-specific LoRA experts during inference, its ability to mitigate forgetting becomes limited, especially when the task gaps are large.

**Evaluation on General Tasks** We evaluate general knowledge retention by comparing the original pretrained MLLM (Zero-shot) and the final models of Seq-FT, O-LoRA, and our proposed D-MoLE after training on all nine tasks. The evaluation uses widely-adopted benchmarks MME (Fu et al., 2024), MMMU (Yue et al., 2024), and POPE (Li et al., 2023), which assess the general abilities of MLLMs. As shown in Table 3, our method outperforms the baselines, showing that it not only mitigates forgetting during the CMIT process but also reduces the loss of general abilities often seen in continual learning scenarios.

## 5.3. Analysis of Training Efficiency

We summarize the total training time of D-MoLEand representative baselines in Table 4. In addition to improved performance, D-MoLE also offers modest gains in training efficiency. This improvement primarily stems from our selective placement of LoRA modules: instead of inserting LoRA into all layers, as done in O-LoRA or joint-learning baselines, our method activates LoRA only in the most sensitive transformer layers. This reduces the number of train-

Table 4: Total training time of our method compared to representative baselines.

| Method | Total Time |
|---|---|
| Joint-learning | 12.83 h |
| Seq-FT | 13.15 h |
| O-LoRA | 14.87 h |
| MoLA | 23.03 h |
| **D-MoLE** | **12.40 h** |

able parameters and backpropagation computations while maintaining adaptation capacity. The full training time comparison can be found in Appendix H.

### 5.4. Ablation Study

To verify the effectiveness of each component, we compare different variants of D-MoLE on our CMIT benchmark:

- **v1**: Only the LLM is fine-tuned, with the vision encoder frozen, to assess the impact of textual modality updates.

- **v2**: Only the vision encoder is fine-tuned, with the LLM frozen, to assess the impact of visual modality updates.

- **v3**: The gradient-based inter-modal continual curriculum is removed, treating textual and visual modalities uniformly without dynamic update adjustments.

- **v4**: The dynamic layer-wise expert allocator is removed, and LoRA experts are added uniformly across layers without task-specific routing.

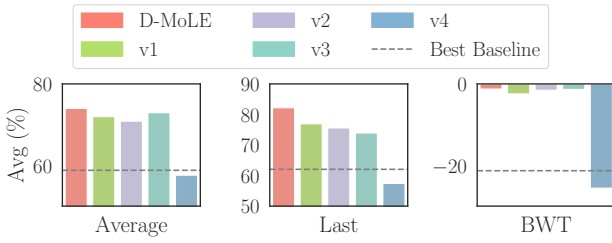

Figure 4: Results of ablation experiments.

The results in Figure 4 highlight the importance of each component in D-MoLE. First, fine-tuning only the LLM (**v1**) or the vision encoder (**v2**) leads to suboptimal results, as MLLMs cannot fully utilize new task information when only one modality is updated. Next, training both without the modality balancing mechanism (**v3**) shows slight improvement over **v1** and **v2**, but still falls short of our full method. This indicates that balanced adaptation between modalities is essential for optimal performance. Finally, without dynamic LoRA expert allocation (**v4**), performance drops significantly across all metrics, underscoring the role of task-specific expert assignment in mitigating forgetting.

## 6. Related Work

**Continual Learning for LLMs and MLLMs**   Recent research in CL for LLMs and MLLMs (Yang et al., 2024; Zheng et al., 2025) focuses on PEFT (Ding et al., 2023). Some works (Wang et al., 2022b;a; Razdaibiedina et al., 2023; Smith et al., 2023; Zhao et al., 2024) introduce query-key prompt pools for task adaptation. However, prompt tuning reduces context length, critical for MLLMs due to extensive token usage. Existing solutions address distinct challenges: Model Tailor (Zhu et al., 2024) focuses on the pretrained knowledge forgetting of MLLM after fine-tuning specific tasks, which is different from our setting of CMIT. O-LoRA (Wang et al., 2023a) employs orthogonal regularization for LoRA modules Eproj (He et al., 2023) allocates task-specific multimodal projectors, Fwd-Prompt (Zheng et al., 2024) constrains gradient interference via residual projections, LLaCA (Qiao et al., 2024) leverages EMA-based updates, and Continual LLaVA (Cao et al., 2024) constructs dual instruction embeddings. Unlike these methods, our approach focuses on architectural evolution to address task architecture conflicts and modality imbalances in CMIT.

**Mixture of LoRA Experts**   Recent research extends the Mixture of Experts (MoE) framework (Jacobs et al., 1991; Shazeer et al., 2017) by integrating LoRA (Hu et al., 2022) as experts within PEFT. MoLE (Wu et al., 2024c) implements layer-wise experts with learned gating, MoA (Yu et al., 2024) uses MoLE to mitigate forgetting in VLM's incremental learning, and LoraMoE (Dou et al., 2024) decouples experts for knowledge and tasks. MixLoRA (Shen et al., 2024) enables input-specific matrix reconfiguration, CoIN (Chen et al., 2024a) employs dense MoELoRA to address forgetting in CMIT, and SMoLoRA (Wang et al., 2024c) introduces separable routing between visual understanding and instruction following. Optimizing expert allocation is another key focus. MoLA (Gao et al., 2024) explores layer-wise allocation strategies, PMoE (Jung & Kim, 2024) concentrates experts in higher layers, and AlphaLoRA (Qing et al., 2024) assigns experts based on layer-specific training quality. Our work focuses on dynamic expert allocation for CMIT, enhancing task adaptability while keeping trainable parameters constrained.

## 7. Conclusion

In this paper, we introduced D-MoLE, a novel framework for CMIT that dynamically allocates LoRA experts across layers using zero-cost metrics. Our method efficiently adapts to new tasks while preserving previously learned knowledge and mitigating modality imbalance through a gradient-based inter-modal curriculum. Experiments show that our method outperforms existing approaches in task adaptation and knowledge retention, providing a scalable and efficient solution for continual learning in MLLMs.

## Acknowledgement

This work is supported by National Natural Science Foundation of China No.62222209, Beijing National Research Center for Information Science and Technology under Grant No.BNR2023TD03006.

## Impact Statement

This paper presents work whose goal is to advance the field of Machine Learning. There are many potential societal consequences of our work, none which we feel must be specifically highlighted here.

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

# Appendix of D-MoLE: Dynamic Mixture of Curriculum LoRA Experts for Continual Multimodal Instruction Tuning

This appendix provides supplementary materials that complement the main text by presenting additional analyses, implementation details, and discussions:

## A. Implementation Details

We conduct our experiments using the InternVL2-2B model (Chen et al., 2024b) on eight NVIDIA A100 GPUs. We use bfloat16 precision for training to improve computational efficiency and performance on the A100 GPUs. LoRA is applied to the attention layers $W^{qkv}$ and the projection layers $W^o$ in each transformer block. To ensure a fair comparison, we maintain an equal number of trainable parameters across all baselines and our method. Specifically, we use a rank of 4 for the Seq-FT baseline and a rank of 8 for our method, with an allocation budget ratio of 0.5. Since both the LLM and the vision encoder in our base model have 24 layers, the total parameter budget $B_{\text{total}}$ is set to $(24 + 24) \times 0.5 = 24$.

The learning rate is set to 1e-4, with 5 training epochs for the SK-VG and VizWiz-VQA datasets, and 1 epoch for the other datasets. The batch size per device is 10 for most datasets, except for the OCR-VQA dataset, which uses a batch size of 8. To reduce the number of training samples in the OCR-VQA dataset, we consolidate QA pairs for the same image into multi-round conversations. The image feature dimension $d_v$ is 1024, and the text feature dimension $d_t$ is 2048, resulting in an input and output dimension of 3072 for the autoencoder, with a hidden dimension of 128. Each autoencoder is trained for 100 epochs with a learning rate of 1e-3.

## B. Additional Related Work

**Multimodal Large Language Models**   Multimodal large language models (Dai et al., 2023; McKinzie et al., 2024; Zhu et al., 2025; Bai et al., 2025) have attracted attention for their ability to handle text along with other modalities, such as images and videos. By combining LLMs with multimodal encoders, they support tasks such as image captioning and visual question answering. Pretraining MLLMs requires extensive data and computation, often taking weeks or months. As new tasks and datasets emerge, some models (Liu et al., 2023; Chen et al., 2024b) suggest mixing new task data with pretraining data to preserve generalization, though this is computationally expensive and sometimes infeasible. Our work focuses on continually adapting pretrained MLLMs to new tasks without relying on data replay.

**Traditional Continual Learning Methods**  Continual Learning (CL) (De Lange et al., 2021; Wang et al., 2024a) aims to prevent catastrophic forgetting, where models lose the ability to perform previously learned tasks when adapting to new ones. Traditional CL methods include replay-based (Rebuffi et al., 2017; Chaudhry et al., 2019a;b; Rolnick et al., 2019; Lopez-Paz & Ranzato, 2017), parameter regularization-based (Kirkpatrick et al., 2017; Li & Hoiem, 2017; Lee et al., 2017; Aljundi et al., 2018; Farajtabar et al., 2020), and model expansion-based approaches (Serra et al., 2018; Rusu et al., 2016; Mallya & Lazebnik, 2018; Yoon et al., 2018). Replay-based methods store or generate data to retain prior knowledge, while regularization techniques like EWC (Kirkpatrick et al., 2017) apply constraints to important parameters. Model expansion increases capacity to handle new tasks while preserving prior information.

**Mixture of Experts**  The Mixture of Experts (MoE) model, introduced by (Jacobs et al., 1991), uses a gating mechanism to determine each expert's contribution, where all experts are activated, enhancing performance but at high computational cost. Sparsely Gated MoE (SMoE) (Shazeer et al., 2017) improves efficiency by activating only a subset of experts for each input. Switch Transformer (Fedus et al., 2022) and GShard (Lepikhin et al., 2021) further optimize expert activation for large NLP and vision models. Recent research has extended the Mixture of Experts (MoE) framework (Jacobs et al., 1991; Shazeer et al., 2017) by integrating LoRA (Hu et al., 2022) as experts within parameter-efficient fine-tuning (PEFT) frameworks. These models are called Mixture of LoRA Experts (MoLE).

**Curriculum Learning**  Curriculum learning organizes training in an easy-to-hard order to improve optimization and generalization (Wang et al., 2021). It has been applied in recommendation (Chen et al., 2021a;b; Wang et al., 2023c), multimodal grounding and pretraining (Chen et al., 2023a; Lan et al., 2023; Huang et al., 2024; Zhou et al., 2023), neural architecture search (Qin et al., 2023; Zhou et al., 2022; Yao et al., 2024), and structured prediction (Zhang et al., 2022). Recent benchmarks such as CurBench (Zhou et al., 2024) further enable systematic evaluation. Recently, curriculum learning is also widely adopted in LLMs' pretraining process, e.g., Kimi K1.5 (Team et al., 2025), DeepSeek-Prover-V2 (Ren et al., 2025), Seed-Coder. Our work introduces a gradient-based inter-modal curriculum for CMIT.

# C. Detailed Algorithm of D-MoLE

---

**Algorithm 1** Training Process for Continual Learning on Task $\mathcal{T}_t$

---

**Input:** New task $\mathcal{T}_t$, pretrained MLLM, previous task-specific LoRA experts, task data $\mathcal{D}_t$, parameter budgets $B_{\text{LLM}}$, $B_{\text{Vision}}$

**Output:** Updated model with new task-specific experts and task recognition

/* Step 1: Initialization */

$\mathcal{D}_{\text{sub}} \leftarrow \text{RANDOMSAMPLING}(\mathcal{D}_t)$

Compute zero-cost scores for $\mathcal{D}_{\text{sub}}$ as in Eq. 3

/* Step 2: Modality Update and Budget Allocation */

Compute update ratio for LLM and vision encoder based on task difficulty (Eq. 14)

Allocate budget for each modality as in Eq. 15

/* Step 3: Expert Allocation */

Rank layers by gradient sensitivity (Eq. 11)

Allocate LoRA experts to top layers based on the budget (Eq. 5)

/* Step 4: Autoencoder Training */

Extract pooled embeddings from $\mathcal{D}_{\text{sub}}$ (Eq. 7)

Train autoencoder for task $\mathcal{T}_t$ using MSE loss; set threshold $\tau_t$

/* Step 5: Training with Task-Specific Experts */

Freeze pretrained weights and previous LoRA experts

Train newly allocated experts on $\mathcal{D}_t$

/* Step 6: Evaluation */

For each sample, use autoencoders for expert selection and routing

If classified as unknown, process sample with pretrained MLLM

---

Algorithm 1 provides the detailed pipeline of D-MoLE for continual learning on task $\mathcal{T}_t$, as described in Section 4.3.

The process starts with zero-cost proxies identifying the most critical layers for adaptation. A gradient-based inter-modal curriculum then computes the optimal update ratio between the LLM and vision encoder. LoRA experts are dynamically

allocated across layers based on modality-specific parameter budgets. The model is fine-tuned on the new task, updating only the newly introduced LoRA experts. During evaluation, task-specific autoencoders manage expert routing and task recognition to ensure accurate performance across tasks.

## D. Notations

Table 5: The summary of the notations and their descriptions.

| Notations | Descriptions |
|---|---|
| $\mathcal{T}_t$ | The $t$-th task in CMIT |
| $\mathcal{D}_t$ | Training data subset for task $\mathcal{T}_t$ |
| $\mathcal{D}_{\text{sub},t}$ | Data subset for zero-cost score computation |
| $B_{\text{total}}$ | Total parameter budget for each task |
| $B_{\text{M}}^t$ | LoRA experts for module M in task $\mathcal{T}_t$ |
| $r_{\text{M}}^t$ | Ratio of LoRA experts for module M |
| M | Module (either LLM or Vision encoder) |
| $\nabla W_l^t$ | Gradient for layer $l$ during task $\mathcal{T}_t$ |
| $\|\nabla W_l^t\|_2$ | $L_2$-norm of the gradient for layer $l$ |
| $\mathcal{R}$ | Set of relevant tasks based on reconstruction loss |
| $\tau_t$ | Reconstruction loss threshold |
| $\text{Rank}(t)$ | Rank of task $t$ based on reconstruction loss |
| $g_l^k(x)$ | Gating function for expert $k$ at layer $l$ |
| $I_l^t$ | LoRA expert allocation indicator for layer $l$ |
| $\mathbb{I}$ | Characteristic function, outputs 1 if condition is met |
| $v_{\text{pooled}}$ | Pooled image features |
| $w_{\text{pooled}}$ | Pooled text features |
| $z$ | Concatenated image and text features |
| concat | Concatenation function for image and text features |
| $\hat{z}^t$ | Reconstructed output from autoencoder for task $\mathcal{T}_t$ |
| $\mathcal{L}_{\text{rec}}^t(z)$ | Reconstruction loss for input $z$ |
| $\mathcal{L}_{\text{MSE}}$ | MSE loss function |
| $\mathcal{L}(\mathcal{D}_{\text{sub},t})$ | Loss on subset $\mathcal{D}_{\text{sub},t}$ |
| $\text{Score}_{\text{M}}^t$ | Difficulty score for module M |
| TopK | Function selecting top-k tasks by rank |

## E. Construction Details of the CMIT Benchmark

Our CMIT benchmark is designed to comprehensively evaluate the model's ability to adapt to diverse multimodal inputs and varying task complexities. The details of the datasets, instruction templates, and the construction of the data sequence for continual learning experiments are described below.

**Dataset Diversity**  The datasets in our benchmark include a wide range of data types, such as natural images, text, diagrams, and medical images, ensuring coverage across various real-world scenarios. These datasets also encompass tasks of differing complexity, including diagram reasoning in IconQA and scene grounding in SK-VG, enabling a thorough evaluation of both generalization and knowledge retention capabilities.

**Instruction Templates**  The instruction templates used in these datasets are provided in Table 6. To maintain consistency, we adopt a unified instruction format across all tasks, avoiding task-specific templates. For tasks with the same output format, we use uniform templates to ensure fair evaluation. This standardization prevents instruction types from serving as implicit task identifiers, allowing performance to depend solely on task content and enabling a more accurate evaluation of the model's ability to mitigate catastrophic forgetting.

**Task Sequence Construction**  During our experiments, the sequence of datasets is randomized, alternating between different task types to create a more challenging and dynamic environment. This setup rigorously tests the model's

Table 6: Instruction templates for each dataset used in the CMIT Benchmark.

| Task Type | Dataset Name | Instruction Template |
|---|---|---|
| **Open-ended VQA** | VizWiz-VQA (Gurari et al., 2018)
OCR-VQA (Mishra et al., 2019)
KVQA (Shah et al., 2019) | `<image>`
`[QUESTION]`
Answer the question with a single word or phrase. |
| **Multi-choice VQA** | IconQA (Lu et al., 2021)
PMC-VQA (Zhang et al., 2024a) | `<image>`
`[QUESTION]`
Options: A. `[Choice_1]`, B. `[Choice_2]`, ...
Answer with the option's letter directly. |
| **Image Captioning** | VizWiz-Caption (Gurari et al., 2020)
TextCaps (Sidorov et al., 2020)
Flickr30k (Young et al., 2014) | `<image>`
Provide a one-sentence caption for the provided image. |
| **Visual Grounding** | SK-VG (Chen et al., 2023b) | `<image>`
Based on the Knowledge: `[KNOWLEDGE]`
Please provide the bounding box coordinates of the region `<ref>[DESCRIPTION]</ref>`. |

adaptability and its capacity to maintain performance on previous tasks while learning new ones.

## F. Analysis of Antoencoder Router in D-MoLE

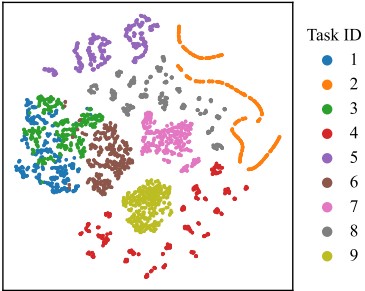

Figure 5: t-sne of the reconstructed sequence embeddings of the task-specific autoencoders.

As described in Section 4.1, we use task-specific autoencoders as routers to capture and differentiate sequence embeddings for each task. Figure 5 presents a t-SNE visualization of the reconstructed sequence embeddings from each task-specific autoencoder. To generate this figure, we randomly sample 500 training instances from each task and extract their multimodal instruction sequence embeddings following the procedure in Section 4.1. We then reconstruct these embeddings using their respective autoencoders and project them into 2D space using t-SNE.

The resulting visualization shows clear separation among tasks, suggesting that even a simple 2-layer autoencoder (one encoder and one decoder) is sufficient to learn discriminative representations. While some captioning tasks are relatively close, they remain largely distinguishable, indicating that the autoencoders successfully capture subtle differences between similar tasks.

We adopt this lightweight architecture to minimize routing overhead. This design choice is supported by the fact that tasks in CMIT are typically defined by dataset boundaries, with significant differences in visual domains, instruction formats, or linguistic styles. In such cases, the autoencoder can effectively distinguish tasks without requiring deep architectures. Extending our approach to more challenging settings with blurred task boundaries may be an interesting direction for future work.

## G. Threshold Selection and Sensitivity

Table 7: Performance of the final checkpoint of D-MoLE under different scaling factors applied to all thresholds $\{\tau_t\}$, where $1\times$ corresponds to the default setting used in the main results. The best result for each column is highlighted in **bold**.

| Scaling Factor | VizWiz-Cap | SK-VG | TextCaps | IconQA | OCR-VQA | Flickr30k | VizWiz-VQA | KVQA | PMC-VQA | *Avg. Last* |
|---|---|---|---|---|---|---|---|---|---|---|
| 0.1× | 148.63 | 61.65 | 127.19 | 92.84 | 53.75 | 78.02 | **69.16** | 43.77 | 48.35 | 80.37 |
| 0.5× | 148.65 | **61.70** | 129.43 | 92.67 | 60.00 | 78.02 | **69.16** | 44.25 | **48.40** | 81.36 |
| 1× | **148.77** | 61.60 | **131.41** | **93.05** | **62.85** | **78.32** | **69.16** | 46.05 | **48.40** | **82.18** |
| 2× | 148.75 | 60.70 | 125.83 | 92.92 | 60.73 | 78.18 | **69.16** | **47.16** | **48.40** | 81.31 |
| 10× | 148.69 | 59.45 | 125.70 | 92.19 | 57.79 | 78.02 | **69.16** | 44.69 | **48.40** | 80.34 |

In our routing mechanism, the threshold in Equation 10 is determined based on the reconstruction loss distribution of each autoencoder over its corresponding task's training set. It is set moderately above the observed loss range to account for minor distribution shifts during evaluation and to allow transferable samples from other tasks to activate relevant experts.

Empirically, we observe that the loss distributions are concentrated with few outliers, and a shared thresholding strategy works across tasks without the need for task-specific tuning. In-task and out-of-task losses are typically well-separated, which is further supported by the t-SNE visualization in Figure 5.

The threshold serves two roles: (1) preventing unrelated experts from being selected purely based on top-2 ranking, and (2) enabling rejection of samples from unseen tasks. Thus, it functions as a coarse filter rather than a precise routing controller, allowing for stable performance even with approximate threshold values.

To examine sensitivity, we evaluate the final checkpoint of D-MoLE under different scaling factors applied to the thresholds during evaluation. The results, shown in Table 7, indicate that average performance remains stable across a wide range of threshold values, demonstrating that D-MoLE is not sensitive to the exact threshold setting as long as it remains within a reasonable range. Note that these experiments do not involve retraining under new thresholds, so slight variations in performance may result from mismatched expert usage patterns during inference.

## H. Comprehensive Analysis of Computational Efficiency

This section provides an analysis of the training time and the computational overhead introduced by proxy computation and dynamic expert routing.

Table 8: Training time for all methods across different datasets (in hours).

| Method | VizWiz-Cap | SK-VG | TextCaps | IconQA | OCR-VQA | Flickr30k | VizWiz-VQA | KVQA | PMC-VQA | *Total* |
|---|---|---|---|---|---|---|---|---|---|---|
| Joint-learning | - | - | - | - | - | - | - | - | - | 12.83 |
| Seq-FT | 1.45 | **0.73** | 1.30 | 0.97 | 2.07 | 1.72 | 1.20 | 1.70 | 1.93 | 13.15 |
| LwF-LoRA | 1.45 | 1.02 | 1.77 | 1.38 | 2.82 | 2.35 | 1.67 | 2.33 | 2.67 | 17.50 |
| EWC-LoRA | 1.45 | 0.83 | 1.35 | 1.03 | 2.15 | 1.78 | 1.28 | 1.77 | 2.02 | 13.73 |
| Dense MoLE | 1.90 | 0.98 | 1.75 | 1.30 | 2.78 | 2.33 | 1.63 | 2.32 | 2.60 | 17.68 |
| Sparse MoLE | 2.60 | 1.37 | 0.87 | 1.78 | 3.73 | 3.88 | 2.20 | 3.17 | 3.45 | 23.07 |
| MoLA | 2.57 | 1.30 | 2.28 | 1.67 | 3.63 | 3.03 | 2.12 | 3.02 | 3.37 | 23.03 |
| O-LoRA | 1.45 | 0.87 | 1.48 | 1.13 | 2.35 | 1.97 | 1.42 | 1.95 | 2.20 | 14.87 |
| **D-MoLE** | **1.28** | **0.73** | **1.20** | **0.93** | **1.98** | **1.62** | **1.15** | **1.62** | **1.85** | **12.40** |

### H.1. Training Time Analysis

Based on the training logs from our experiments, we have compiled the training time for each method, as shown in Table 8. To ensure a fair comparison, all methods use the same number of trainable parameters, and all experiments are conducted on $8 \times$ NVIDIA A100-SXM4-40GB GPUs.

From the results, we observe that although computational efficiency is not the primary focus of our work, our method demonstrates a slight advantage over all baselines, including vanilla LoRA Seq-FT. This advantage stems from our dynamic allocation strategy, which selectively assigns LoRA modules to the most critical layers, reducing computational latency compared to methods that apply LoRA uniformly across all layers. Additionally, our method does not introduce complex auxiliary losses like O-LoRA.

### H.2. Computational Cost of Proxy Computation and Dynamic Expert Routing

In Table 9, we report the time cost for preprocessing (computing the zero-cost proxy scores), training, and their ratio across tasks. The zero-cost proxy computation involves calculating the loss, performing a backward pass to obtain gradients, and computing the gradient norm across layers using a random 1% subset of the training data. Unlike training, this computation uses only a small subset of data and does not update model parameters, resulting in lower computational time. The SK-VG, IconQA, and VizWiz-VQA tasks have lower ratios because they are trained for 5 epochs, while the zero-cost proxy scores are computed only once.

Table 9: Preprocessing time, training time, and their ratio across tasks.

| Method | VizWiz-Cap | SK-VG | TextCaps | IconQA | OCR-VQA | Flickr30k | VizWiz-VQA | KVQA | PMC-VQA | *Total* |
|---|---|---|---|---|---|---|---|---|---|---|
| Preprocessing (s) | 88 | 8 | 79 | 10 | 121 | 100 | 14 | 104 | 126 | 650 |
| Training (s) | 4623 | 2712 | 4377 | 3394 | 7157 | 5849 | 4179 | 5824 | 6657 | 44772 |
| Ratio | 1.91% | 0.29% | 1.81% | 0.31% | 1.69% | 1.71% | 0.34% | 1.79% | 1.90% | 1.45% |

For the computational cost of dynamic expert routing, the process involves two main steps: (1) computing the sequence representation using the LLM and the vision encoder in the MLLM, and (2) computing the reconstruction loss using task-specific autoencoders, which are simple two-layer MLPs with an encoder and a decoder layer.

We evaluate the final model of D-MoLE on 1000 random test samples from the PMC-VQA dataset. The PMC-VQA task is a multiple-choice VQA task, where the model only needs to output the selected option (i.e., a single letter). This makes the inference cost of PMC-VQA the lowest among all tasks in our experiments. Despite this minimal inference cost, the results in Table 10 show that the expert routing time accounts for only a small fraction of the total inference time, demonstrating the efficiency of our dynamic expert routing process.

Table 10: Expert routing time, inference time, and their ratio on PMC-VQA task.

| Stage | Average ± Std (s) |
|---|---|
| Expert Routing Time | 0.0552 ± 0.0035 |
| Inference Time | 0.5225 ± 0.0943 |
| Ratio | 10.56% |

In summary, the computational overhead of using a subset to compute gradients and dynamically routing experts is minimal, with little impact on overall time and efficiency. The results in the tables demonstrate the efficiency of our approach.

## I. Comparison with Baseline at Increased Parameter Budget

In this section, we analyze the performance of D-MoLE in comparison to the most advanced baseline (O-LoRA) under varying trainable parameter budgets. To evaluate the impact of increasing the parameter budget, experiments were conducted with O-LoRA using different rank values ($r$), corresponding to $1\times$, $2\times$, and $4\times$ the default parameter size. The results are summarized in Table 11.

Table 11: Performance comparison of D-MoLE and O-LoRA under varying trainable parameter budgets.

| Method | Trainable Params. | *Average* | *Last* | *BWT* |
|---|---|---|---|---|
| O-LoRA ($r = 4$) | $1\times$ | 58.79 | 62.04 | -21.31 |
| O-LoRA ($r = 8$) | $2\times$ | 60.09 | 59.68 | -19.95 |
| O-LoRA ($r = 16$) | $4\times$ | 59.75 | 59.34 | -20.95 |
| **D-MoLE** | $1\times$ | **73.87** | **82.18** | **-1.49** |

As shown in Table 11, increasing the number of trainable parameters slightly improves the performance of O-LoRA. However, D-MoLE consistently outperforms O-LoRA across all metrics, even with smaller parameter budgets. This suggests that D-MoLE adapts better to new tasks without significant loss of previously learned knowledge.

## J. Visualization of the Dynamic Training Process

To illustrate the adaptability of D-MoLE throughout continual training, we provide visualizations that capture its dynamic behavior in terms of both architecture evolution and expert utilization.

- **Architecture Evolution Dynamics**: As shown in Figure 6, we plot a heatmap that depicts how task-specific LoRA experts are progressively allocated across different transformer layers during the CMIT process. This reflects the evolution of architecture over time based on the changing sensitivity patterns of tasks.

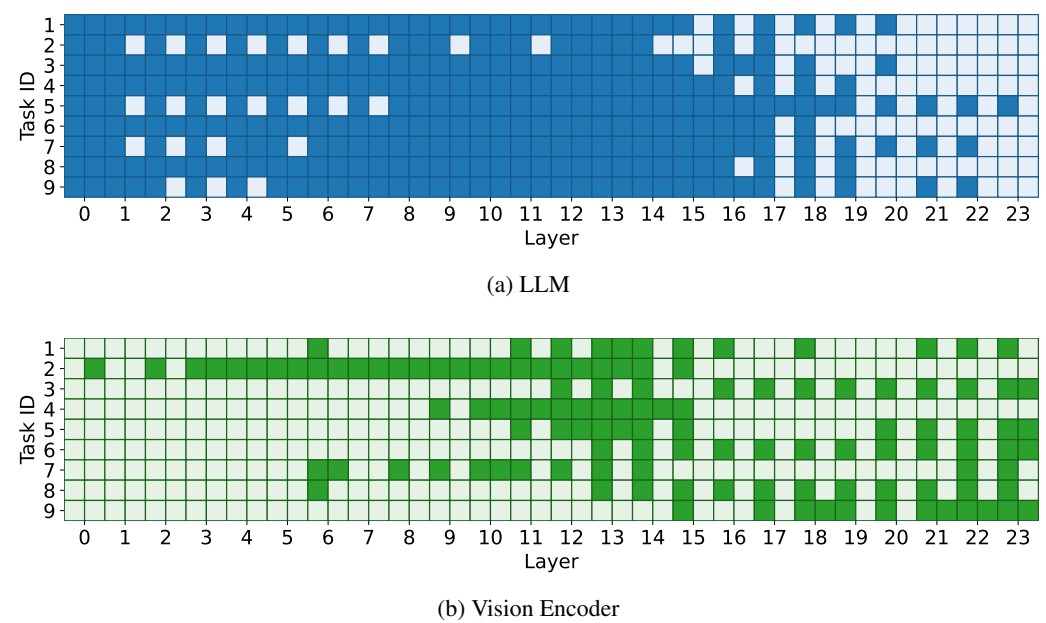

(a) LLM

(b) Vision Encoder

Figure 6: Architecture evolution dynamics during continual multimodal instruction tuning. Each transformer layer is divided into two subslots, corresponding to LoRA applied on attention weights $W_{qkv}$ and projection weights $W_o$. Darker cells indicate that a LoRA expert is allocated to the corresponding position. The order of the task is the same as in Table 2.

- **Expert Activation Dynamics**: Figure 7 presents a heatmap of expert activation across training steps, highlighting how different task-specific experts are selectively routed as the model encounters new tasks. This demonstrates the effectiveness of the autoencoder-based gating mechanism in adapting routing decisions over time.

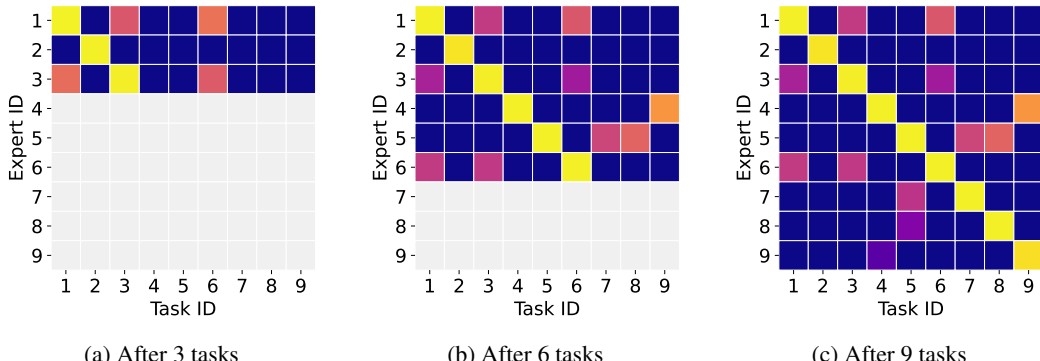

(a) After 3 tasks      (b) After 6 tasks      (c) After 9 tasks

Figure 7: Expert activation dynamics at different stages of continual multimodal instruction tuning. Each cell indicates how frequently a task activates a given expert. Grayed cells correspond to experts which tasks are unseen. Lighter colors denote higher activation frequency. The order of the task is the same as in Table 2.

These visualizations clearly show that D-MoLE does not rely on a static configuration, but continuously adapts expert allocation and usage as training progresses.

## K. Theoretical Analysis of Task Architecture Conflict in CMIT

In this section, we provide a theoretical foundation for the challenge of *task architecture conflict* in continual multimodal instruction tuning (CMIT), which is described in Section 3.1. Specifically, we show that different tasks induce discrepancies in the gradient norms of specific layers, leading to conflicts during sequential training.

We formalize the conflict through the following theorem:

**Theorem K.1** (Theorem 3.1). *Consider an MLLM with $L$ transformer layers trained sequentially on two distinct tasks $\mathcal{T}_A$ and $\mathcal{T}_B$. Under Assumption K.2 (task heterogeneity) and Assumption K.3 (non-collinear gradients), there exists at least one layer $l^* \in \{1, \ldots, L\}$ where the expected gradient norms differ:*

$$\left\| \mathbb{E}_{\mathcal{T}_A} \left[ \nabla_{\mathbf{W}_{l^*}} \mathcal{L} \right] \right\|_2 \neq \left\| \mathbb{E}_{\mathcal{T}_B} \left[ \nabla_{\mathbf{W}_{l^*}} \mathcal{L} \right] \right\|_2, \tag{19}$$

*where $\nabla_{\mathbf{W}_l} \mathcal{L}$ denotes the gradient of loss $\mathcal{L}$ with respect to parameters $\mathbf{W}_l$ at layer $l$.*

Our analysis relies on two foundational assumptions about task properties:

**Assumption K.2** (Task Heterogeneity). *Tasks $\mathcal{T}_A$ and $\mathcal{T}_B$ have distinct input-output distributions ($p_{\mathcal{T}_A}(x, y) \neq p_{\mathcal{T}_B}(x, y)$) and require different optimal parameter configurations.*

**Assumption K.3** (Non-Collinear Expected Gradients). *For at least one layer $l$, the expected gradients of $\mathcal{T}_A$ and $\mathcal{T}_B$ cannot be aligned by scalar scaling:*

$$\forall \alpha \in \mathbb{R}, \quad \mathbb{E}_{\mathcal{T}_A} \left[ \nabla_{\mathbf{W}_l} \mathcal{L} \right] \neq \alpha \cdot \mathbb{E}_{\mathcal{T}_B} \left[ \nabla_{\mathbf{W}_l} \mathcal{L} \right]. \tag{20}$$

We formally define task-specific gradient magnitude as follows:

**Definition K.4** (Gradient Norm of a Layer). *For task $\mathcal{T}$, the gradient norm at layer $l$ is defined as:*

$$G(l, \mathcal{T}) \triangleq \left\| \mathbb{E}_{\mathcal{T}} \left[ \nabla_{\mathbf{W}_l} \mathcal{L} \right] \right\|_2, \tag{21}$$

*where $\| \cdot \|_2$ denotes the L2-norm.*

*Proof of Theorem 3.1.* Assume for contradiction that all layers exhibit equal gradient norms across tasks:

$$\forall l \in \{1, \dots, L\}, \quad G(l, \mathcal{T}_A) = G(l, \mathcal{T}_B). \tag{22}$$

By Assumption K.2, the distinct input-output distributions of $\mathcal{T}_A$ and $\mathcal{T}_B$ necessitate different optimal parameter configurations. This implies the existence of at least one critical layer $l^*$ where their expected gradients differ fundamentally:

$$\mathbb{E}_{\mathcal{T}_A}\left[\nabla_{\mathbf{W}_{l^*}}\mathcal{L}\right] \neq \mathbb{E}_{\mathcal{T}_B}\left[\nabla_{\mathbf{W}_{l^*}}\mathcal{L}\right]. \tag{23}$$

Furthermore, Assumption K.3 specifies that these gradients cannot be related by scalar scaling. Formally, there exists no $\alpha \in \mathbb{R}$ such that:

$$\mathbb{E}_{\mathcal{T}_A}\left[\nabla_{\mathbf{W}_{l^*}}\mathcal{L}\right] = \alpha \cdot \mathbb{E}_{\mathcal{T}_B}\left[\nabla_{\mathbf{W}_{l^*}}\mathcal{L}\right], \tag{24}$$

indicating directional divergence beyond mere magnitude differences.

To analyze this conflict, let $\mathbf{g}_A \coloneqq \mathbb{E}_{\mathcal{T}_A}[\nabla_{\mathbf{W}_{l^*}}\mathcal{L}]$ and $\mathbf{g}_B \coloneqq \mathbb{E}_{\mathcal{T}_B}[\nabla_{\mathbf{W}_{l^*}}\mathcal{L}]$. The initial assumption Eq. (22) enforces equal gradient norms:

$$\|\mathbf{g}_A\|_2 = \|\mathbf{g}_B\|_2. \tag{25}$$

However, the non-collinearity condition Eq. (24) implies $\mathbf{g}_A \not\parallel \mathbf{g}_B$. Geometrically, this requires both gradients to lie on the surface of a common hypersphere while pointing in divergent directions:

$$\mathbf{g}_A, \mathbf{g}_B \in \mathbb{S}^{d-1}(r) \quad \text{with} \quad r = \|\mathbf{g}_A\|_2, \tag{26}$$

where $\mathbb{S}^{d-1}(r)$ denotes a $(d-1)$-dimensional hypersphere of radius $r$.

This geometric constraint creates an optimization dilemma. Sequential training on $\mathcal{T}_A$ and $\mathcal{T}_B$ would demand opposing parameter updates at layer $l^*$:

$$\begin{cases} \mathbf{W}_{l^*} \leftarrow \mathbf{W}_{l^*} - \eta \mathbf{g}_A & (\text{Task } \mathcal{T}_A) \\ \mathbf{W}_{l^*} \leftarrow \mathbf{W}_{l^*} - \eta \mathbf{g}_B & (\text{Task } \mathcal{T}_B) \end{cases} \tag{27}$$

where $\eta$ is the learning rate. Since $\mathbf{g}_A$ and $\mathbf{g}_B$ share equal norms but conflicting directions, these updates attempt to steer $\mathbf{W}_{l^*}$ toward incompatible regions of the parameter space.

Maintaining equal gradient norms across all layers under Eq. (22) would necessitate reconciling these incompatible updates—a requirement that violates the sequential training paradigm. Consequently, the initial assumption cannot hold, thereby proving the existence of at least one layer $l^*$ with $\|\mathbf{g}_A\|_2 \neq \|\mathbf{g}_B\|_2$. $\qquad \square$

*Remark K.5.* Theorem 3.1 highlights the challenge of *task architecture conflict* in CMIT, where sequential task training results in gradient norm discrepancies at certain layers. This conflict indicates that task-specific adaptations demand distinct parameter updates. D-MoLEmitigates this issue by dynamically assigning LoRA experts to layers with higher task sensitivity, promoting improved task adaptation during CMIT.

