# OpenReview forum: "Dynamic Mixture of Curriculum LoRA Experts for Continual Multimodal Instruction Tuning"
_ICML.cc/2025/Conference — ICML 2025 poster_

### Official Review · Reviewer_xQiS · 2025-03-13

**Overall Recommendation:** 4

**Summary:**

This paper presents an algorithm D-MoLE for continual multimodal instruction tuning. The algorithm solves the challenges of task architecture conflict and modality imbalance by dynamically assigning LoRA experts and a gradient-based continual curriculum. Experimental results show the effectiveness of the proposed algorithm.

**Claims And Evidence:**

Yes

**Essential References Not Discussed:**

No

**Experimental Designs Or Analyses:**

Yes

**Methods And Evaluation Criteria:**

Yes

**Other Comments Or Suggestions:**

see above weaknesses

**Other Strengths And Weaknesses:**

Strengths:

1. The paper has the theoretical analysis

2. The setting the paper focuses on is significant

3. The writing is good and easy to read.

Weaknesses:

1. "fixed architecture models inevitably face the dilemma..." mentioned in paragraph 2 of the Introduction is not unique to MLLMs. In fact, continual learning has been extensively studied on this dilemma. The description here is confusing.

2. How to select the threshold hyperparameter of equation 9 and its sensitivity should be discussed.

3. The experimental conditions of figure 5 in Appendix G are not explained. Also, is the use of a 2-layer mlp sufficient to distinguish between different tasks, and is this related to the task difficulty of the dataset itself?

**Questions For Authors:**

see above weaknesses

**Relation To Broader Scientific Literature:**

With the popularity of large-scale pre-training, continual multimodal instruction tuning has become an important topic. This paper focuses on it, which is helpful for the development of multimodal large language models.

**Theoretical Claims:**

Yes

---

> ### Author Rebuttal · Authors · 2025-03-30
>
> We sincerely thank the reviewer for the helpful and encouraging comments. We appreciate that they note the inclusion of theoretical analysis, find our writing clear and easy to read, and recognize the significance of the problem setting. We hope the following explanations provide sufficient clarification.
>
> ---
> **Q1. Tasks-Architecture Conflict in MLLMs**
>
> Thank you for the question. Our intention here is to emphasize why fixed architectures pose additional challenges for continual learning (CL) in MLLMs. We would like to clarify it further here.
>
> MLLMs have inherent architectural heterogeneity, especially in the commonly adopted vision encoder + projector + LLM framework. Unlike unimodal LLMs, MLLMs process inputs from multiple modalities. As tasks vary in their reliance on each modality, abstraction requirements differ across modules. This leads to greater task-wise architectural sensitivity during CL. As shown in Figure 1, the sensitivity of transformer layers in both the vision encoder and the LLM varies substantially across tasks, indicating that different components must adapt differently. This observation motivates us to move beyond fixed architectures and adopt dynamic expert allocation, and further supports our design of an architecture evolution mechanism tailored to the evolving demands of CL in MLLMs.
>
> We will revise the Introduction to better explain this motivation.
>
> ---
> **Q2. Threshold Selection and Sensitivity**
>
> Thank you for the question. The threshold in Equation 9 is selected based on the reconstruction loss distribution of each autoencoder on its corresponding task’s training data. It is set moderately above the typical loss range to accommodate minor distribution shifts at evaluation and to allow transferable samples from other tasks to activate relevant experts.
>
> We observe that the loss distributions are concentrated with few outliers, and the same thresholding strategy works across tasks without tuning. Moreover, in-task and out-of-task losses are typically well-separated, as supported by the t-SNE visualization in Figure 5.
>
> The threshold serves two purposes: (1) avoiding irrelevant experts from being activated solely due to top-2 ranking, and (2) enabling detection of unseen tasks. As such, it plays a filtering role rather than directly driving routing decisions. This allows for coarse-grained threshold choices without requiring precise tuning.
>
> To assess sensitivity, we evaluate the final checkpoint of D-MoLE under different scaling factors of the thresholds $\\{\tau_t\\}$ and report the corresponding average Last scores across tasks, with $1\times$ denoting the default setting. Full table can be found at https://anonymous.4open.science/r/D-MoLE/table10.jpg.
>
> **Table 1: Average task performance under different threshold scaling factors, evaluated on the final checkpoint.**
> |Scaling Factor|$0.1\times$|$0.5\times$|$1\times$|$2\times$|$10\times$|
> |-|-|-|-|-|-|
> |*Avg. Last*|80.37|81.36|**82.18**|81.31|80.34|
>
> Note that this experiment applies threshold scaling only during evaluation. Due to current computational resource constraints, we do not retrain the model under each new threshold setting. As a result, expert collaboration patterns during evaluation may differ from those formed during training under the default thresholds, which may slightly affect performance. Despite this, performance remains stable across a wide range of threshold scaling factors, indicating that our method is not sensitive to the exact threshold values, as long as they are within the same order of magnitude. We will include this analysis in the revised version.
>
> ---
> **Q3. Setup of Figure 5 and Autoencoder Capacity**
>
> Thank you for pointing this out. The experimental setup for Figure 5 (Appendix G) is as follows: we randomly sample 500 training samples from each task and extract their multimodal instruction sequence embeddings using the method described in Section 4.1. We then compute the reconstructed embeddings using the corresponding autoencoders and visualize all embeddings together using t-SNE. The clear separation among clusters provides supporting evidence that task-specific autoencoders learn distinct representations and can effectively function as routers. We will include these experimental details in the revised version for clarity.
>
> Regarding the use of a 2-layer MLP (one encoder and one decoder layer) for the autoencoder, we find it sufficient in our setting. In typical CL benchmarks for MLLMs, tasks are defined by dataset boundaries, and samples from different tasks often differ significantly in terms of image domains, question types, or textual styles. As such, the autoencoders can effectively distinguish tasks even with simple architectures. We adopt this lightweight design to minimize the additional computational overhead introduced by the routing process. Exploring more challenging scenarios with blurred task boundaries may be a promising direction for future work.

---

### Official Review · Reviewer_c9zT · 2025-03-13

**Overall Recommendation:** 4

**Summary:**

The paper presents D-MoLE, a framework for continual multimodal instruction tuning (CMIT) in multimodal large language models (MLLMs). It dynamically allocates LoRA experts across layers using zero-cost metrics and addresses modality imbalance through gradient-based inter-modal curriculum learning. By resolving task architecture conflicts and mitigating modality imbalance, D-MoLE preserves performance on previously learned tasks, achieving a 15% average performance improvement over state-of-the-art baselines.

## update after rebuttal
I have carefully read the authors' rebuttal and the feedback has well addressed my questions and concerns. Thus, I would like to insist on the score of accept. Thanks.

**Claims And Evidence:**

yes.

**Essential References Not Discussed:**

No.

**Experimental Designs Or Analyses:**

yes, all.

**Methods And Evaluation Criteria:**

yes.

**Other Comments Or Suggestions:**

Please see cons.

**Other Strengths And Weaknesses:**

**Pros:**

- The paper provides clear and well-motivated illustrations of the problems and challenges in continual multimodal learning. The issues of task architecture conflict and modality imbalance are effectively addressed by the method's design, particularly through the dynamic expert allocator and modality-specific curriculum.

- The approach is both well-motivated and intuitive. The concept of dynamically adding LoRA parameters is an interesting and potentially practical solution for scaling models while preserving efficiency. This method could have significant real-world applications, especially in resource-constrained environments.

- By resolving task architecture conflicts and mitigating modality imbalance, D-MoLE successfully preserves performance on previously learned tasks, leading to an impressive 15% average improvement over state-of-the-art baselines. This highlights the method’s effectiveness.

- The ablation studies are thorough and provide strong evidence of the effectiveness of each module in the framework. Additionally, supplementary experiments demonstrate that the method does not add many additional computational costs.

**Cons:**

- It would be beneficial to include some case studies or examples of specific tasks to further illustrate the method's performance in different settings and help contextualize its real-world applications.

- Some of the figures could be enhanced for better clarity. For instance, the fonts in Figure 1 are too small, which may hinder readability, especially for audiences engaging with the paper on printed formats.

- Showcasing the dynamic training process in more detail would be helpful. For example, visualizing how experts are assigned throughout the continual learning process would provide a clearer understanding of the method's adaptability.

- Could the proposed method be combined with other continual learning approaches, such as O-LoRA, to further enhance performance?

- Is there a need to remove experts over time, in addition to adding them?

**Questions For Authors:**

Please see cons.

**Relation To Broader Scientific Literature:**

It could be applied in scientific fields related to MLLM, e.g. medical imaging.

**Theoretical Claims:**

yes, theorem 3.1.

---

> ### Author Rebuttal · Authors · 2025-03-30
>
> We sincerely thank the reviewer for the encouraging and constructive feedback. We are pleased that they find our approach well-motivated, intuitive, and effective, and acknowledge the impressive performance gains. We hope our responses below address the remaining concerns.
>
> ---
> **Q1. Including Some Case Studies or Examples**
>
> Thank you for the valuable suggestion. In the revised version, we will include case studies in the appendix. For example, we plan to show some multimodal instruction examples from early tasks and compare the model’s responses to them after training on later tasks. This will help illustrate how well D-MoLE preserves prior knowledge by examining model behavior on early-task examples at different stages of continual learning.
>
> ---
> **Q2. Figure Clarity and Font Size**
>
> Thank you for the helpful suggestion. We will increase the font size in Figure 1 to improve readability, and recheck all figures in the paper to ensure visual clarity in the revised version.
>
> ---
> **Q3. Visualizing the Dynamic Training Process**
>
> Thank you for the valuable suggestion. We have created two heatmaps to illustrate the dynamic behavior of our method and better convey the adaptability of D-MoLE. We provide preview versions at anonymous links below, and will incorporate them into the revised version:
>
> * Architecture evolution dynamics (https://anonymous.4open.science/r/D-MoLE/figure6.jpg): shows how experts are allocated across different layers during the CMIT process.
> * Expert activation dynamics (https://anonymous.4open.science/r/D-MoLE/figure7.jpg): visualizes how task-specific experts are activated over time during training.
>
> ---
> **Q4. Compatibility with Other Continual Learning Approaches**
>
> Thank you for the question. Combining our method with other continual learning approaches is certainly feasible.
>
> Our method focuses on architectural evolution, dynamically expanding model capacity throughout the CMIT process. This enables flexible task adaptation without being constrained by a fixed parameter budget. In contrast, approaches like O-LoRA mitigate forgetting through parameter regularization, but often face a trade-off between retaining prior knowledge and adapting to new tasks due to their fixed capacity.
>
> These two directions are compatible in principle. While our current framework does not include regularization terms in the loss function, incorporating them may be a promising extension. For example, enforcing orthogonality between different experts could help reduce redundancy and improve task separation.
>
> ---
> **Q5. Need for Expert Removal**
>
> Thank you for the question. While our current experimental setting does not involve expert removal, since we evaluate all tasks after each training stage to assess knowledge retention (see Appendix F), our framework can easily support it.
>
> For example, one practical extension is to track expert activation frequency using a sliding window over recent inputs. Experts associated with inactive tasks could then be unloaded to reduce memory and computation overhead, with the option to reload or reinitialize them if needed. This may be a promising direction for future work, particularly in real-world online continual learning scenarios.

---

> > ### Comment · Reviewer_c9zT · 2025-04-02
> >
> > I have carefully read the authors' rebuttal and the feedback has well addressed my questions and concerns. Thus, I would like to insist on the score of 4 (accept) . Thanks.

---

> > > ### Author Response · Authors · 2025-04-02
> > >
> > > Thank you for your kind response. We are glad to hear that our clarifications addressed your concerns. We sincerely appreciate the time and effort you devoted to reviewing our work.

---

### Official Review · Reviewer_cVka · 2025-03-14

**Overall Recommendation:** 4

**Summary:**

This paper addresses the challenge of continual multimodal instruction tuning (CMIT) for Multimodal Large Language Models (MLLMs) by proposing a novel Dynamic Mixture of Curriculum LoRA Experts (D-MoLE) method. Unlike fixed-architecture models that struggle with adapting to new tasks, D-MoLE dynamically evolves the model’s architecture within a parameter budget by allocating LoRA experts layer-wise and adjusting update ratios based on modality difficulty. Experimental results show that D-MoLE outperforms state-of-the-art baselines by 15% on average, making it the first study to tackle continual learning for MLLMs from an architectural perspective.

**Claims And Evidence:**

Yes

**Essential References Not Discussed:**

No

**Experimental Designs Or Analyses:**

Yes

**Methods And Evaluation Criteria:**

Yes

**Other Comments Or Suggestions:**

See weaknesses.

**Other Strengths And Weaknesses:**

**Strengths:**
The paper is well-written and easy to follow, making it accessible to a broad audience. The topic of continual learning for multimodal large language models is highly relevant in contemporary research, with significant real-world applications. The study effectively highlights the challenges of task architecture conflict and modality imbalance, supporting these discussions with both empirical results and theoretical analyses.

I find the experimental results particularly impressive, as they demonstrate a substantial improvement over state-of-the-art baselines. The proposed method is intuitively designed and does not introduce excessive complexity in its implementation, which enhances its practicality and reproducibility. The paper presents a strong contribution to the field, addressing key issues with clear and well-supported findings.

**Weaknesses:**
The presentation of results in Table 2 could be confusing due to the presence of two 'average' metrics—one in the dataset row and another in the method row. It would be helpful to clarify how these averages are computed and their intended interpretation.

The Seq-FT baseline achieves the best performance on KVQA in the 'Last' and 'BWT' metrics. Could you provide an explanation for this result? It seems counterintuitive that the most naive baseline would outperform more sophisticated approaches in these measures.

The proposed D-MoLE method significantly outperforms other baselines on the first task, VizWiz-Cap. Could this be an indication that the model is primarily excelling at fitting the initial dataset, rather than truly demonstrating strong continual learning capabilities? Is there any evidence to rule out the possibility that the method behaves more like fine-tuning rather than an effective continual learning approach?

Could you provide further insight into the key factors driving the performance improvements of your method? Specifically, which aspects of the design contribute most to the observed gains? A more detailed breakdown of the improvements would strengthen the paper’s claims.

**Questions For Authors:**

See weaknesses.

**Relation To Broader Scientific Literature:**

N/A

**Theoretical Claims:**

Yes

---

> ### Author Rebuttal · Authors · 2025-03-30
>
> We sincerely thank the reviewer for the thoughtful and positive feedback. It is encouraging that they find our paper well-written and easy to follow, our results impressive, and our method practical, intuitive, and concise. We hope the clarifications below address the reviewer’s remaining concerns.
>
> ---
> **Q1. Clarification of the Two 'Average' Metrics in Table 2**
>
> Thank you for the valuable suggestion. To clarify:
>
> * The *Average* column (rightmost) shows the mean performance of each method across all datasets (i.e., row-wise average).
> * The **Average** row near the top refers to one of the continual learning (CL) metrics introduced in Section 5.1, alongside Last and BWT. Their formal definitions are provided in Appendix F.
>
> To improve clarity, we will consider renaming the CL-specific **Average** row to **AVG** in the revised version.
>
> ---
> **Q2. Unexpectedly Strong Seq-FT Performance on KVQA**
>
> Thank you for pointing out this observation. Seq-FT achieves relatively high accuracy on KVQA immediately after training and experiences minimal forgetting after training on PMC-VQA. These two factors together explain its higher Last and BWT scores on KVQA.
>
> We attribute this behavior to two possible factors. First, KVQA is a relatively difficult dataset, where most methods show only modest improvements over the zero-shot baseline. CL methods that incorporate regularization mechanisms (e.g., LwF-LoRA, EWC-LoRA) may struggle to fully adapt to such tasks due to limited flexibility, whereas Seq-FT’s unconstrained fine-tuning can more easily fit the data. Second, PMC-VQA is a medical-domain task with different visual domains, question types, and prompt formats compared to KVQA. This reduces representational overlap between the two tasks and thus limits interference during sequential training. However, regularization-based methods may apply global constraints which could result in unintended interference with prior task-specific adaptation. A similar pattern is observed between VizWiz-Cap and SK-VG, where Seq-FT also exhibits reduced forgetting, likely due to the same reason.
>
> Overall, while Seq-FT achieves higher Last and BWT scores on KVQA, the margins are small. Across the full task sequence, it still suffers from severe forgetting, as reflected in its low overall CL scores.
>
> ---
> **Q3. Strong Performance on the Initial Task**
>
> Thank you for the thoughtful question. Our training protocol treats all tasks in the sequence uniformly and does not apply any special treatment to the initial task, so there is no risk that our method simply overfits to the first dataset. As evidenced by the overall experimental results, our method outperforms all state-of-the-art baselines on all three CL metrics across most tasks, not just the initial one.
>
> The seemingly large performance gains on VizWiz-Cap can be explained as follows:
> * Following the CoIN benchmark [1], we use CIDEr as the evaluation metric for captioning tasks. Since CIDEr scores are not upper-bounded (often exceeding 1), numerical improvements may appear larger in magnitude.
> * In continual learning, forgetting accumulates over time and amplifies performance degradation on earlier tasks. As D-MoLE better mitigates forgetting, its advantage is more visible on tasks like VizWiz-Cap that appear early in the sequence.
>
> ---
> **Q4. Key Factors Driving Performance Improvements**
>
> Thank you for the question. The performance gains of D-MoLE stem from the integration of two complementary modules, each addressing a key challenge in CMIT. The dynamic layer-wise expert allocator addresses task architecture conflict by assigning LoRA experts only to the most relevant layers and routing inputs via task-specific autoencoders. This enables precise architectural adaptation and selective expert activation during inference. The gradient-based inter-modal continual curriculum mitigates modality imbalance by adjusting training dynamics when tasks rely unevenly on different modalities, ensuring stable multimodal adaptation.
>
> Our ablation study (Section 5.3) supports this analysis. Removing the expert allocator (v4) leads to the largest performance drop, underscoring the importance of architectural flexibility. Removing the curriculum module (v3) results in lower performance than the full model, demonstrating its complementary benefit. Limiting updates to a single modality (v1 or v2) also yields suboptimal results, highlighting the need for full-modality adaptation.
>
> Overall, while the expert allocator contributes most directly to performance gains, both components are essential and jointly enable robust and balanced continual learning. We will add more detailed discussion of these factors in the revised version.
>
> ---
> **Reference**
>
> [1] CoIN: A Benchmark of Continual Instruction Tuning for Multimodel Large Language Models. NeurIPS 2024.

---

> > ### Comment · Reviewer_cVka · 2025-04-07
> >
> > Thank you for the authors' substantial effort in addressing my concerns. I have also read the discussions between the authors and other reviewers. I will maintain my acceptance of this submission and hope the authors carefully incorporate the suggestions.

---

> > > ### Author Response · Authors · 2025-04-08
> > >
> > > Thank you for your positive feedback and for taking the time to review our responses. We will carefully incorporate the suggestions in the final version.

---

### Official Review · Reviewer_VwXs · 2025-03-15

**Overall Recommendation:** 4

**Summary:**

This paper presents D-MoLE, a  framework designed to tackle the challenges of continual multimodal instruction tuning (CMIT) in Multimodal Large Language Models (MLLMs). D-MoLE employs a dynamic layer-wise expert allocation strategy to overcome task architecture conflicts and a gradient-based inter-modal continual curriculum to address modality imbalances. This approach facilitates adaptive and efficient learning while maintaining a constrained parameter budget.

**Claims And Evidence:**

yes

**Essential References Not Discussed:**

no

**Experimental Designs Or Analyses:**

yes, results of in sec 5.

**Methods And Evaluation Criteria:**

yes

**Other Comments Or Suggestions:**

The notation could be refined to more clearly differentiate between scalars and vectors, ensuring consistency and readability throughout the paper. Using distinct formatting, such as boldface or arrows for vectors and standard italicization for scalars, would enhance clarity and prevent potential ambiguity in mathematical expressions.

**Other Strengths And Weaknesses:**

pros:

1. The paper is well-structured and clearly presents the challenges of continual learning in multimodal large language models. In Section 3, two preliminary studies effectively demonstrate why traditional continual learning methods may not be suitable for MLLMs. The inclusion of theoretical analyses provides deeper insights into the observed empirical phenomena and highlights the necessity of architectural evolution in continual learning.

2. The idea of automatically evolving the model architecture by incorporating LoRA modules into MLLMs during the continual learning process is particularly inspiring. This approach expands the model's capacity to accommodate new tasks while mitigating performance degradation on previously learned tasks. The concept of gradually increasing parameters over time could become an essential strategy for enabling continual learning in large-scale models, and this paper serves as an important step toward that direction.

3. The experimental results are strong, demonstrating substantial performance gains over state-of-the-art baselines. The improvements over the benchmarks verify the effectiveness of the proposed approach and highlight its potential for advancing continual learning in multimodal scenarios.

Cons:

There are no major concerns with the paper. However, there are some minor points that could benefit from clarification—please refer to the suggestions and questions for further details.

**Questions For Authors:**

1. Several continual learning baselines perform significantly worse than fine-tuning on the first task. However, Seq-FT, which appears to be functionally equivalent to fine-tuning in the initial task, shows notably lower performance. Could you clarify the reasons behind this discrepancy? Are there specific factors, such as optimization dynamics or architectural constraints, that might contribute to this difference?

2. In Table 6, the proposed method demonstrates a shorter training time compared to joint learning and O-LoRA. Given that the architecture evolution process introduces additional parameters to be trained, this result seems somewhat counterintuitive. Could you elaborate on why this occurs? Does the efficiency stem from selective parameter updates, optimized training strategies, or some other factors?

3. Regarding the routing mechanism, is there any specific design choice implemented to ensure a balanced load distribution among experts? If so, could you provide details on how the router dynamically manages workload allocation, especially in scenarios with varying modality distributions?

**Relation To Broader Scientific Literature:**

no much related

**Theoretical Claims:**

yes, I checked theorem 3.1.

---

> ### Author Rebuttal · Authors · 2025-03-30
>
> We sincerely thank the reviewer for the thoughtful and encouraging feedback. We are glad that they find our paper well-structured and our approach to be inspiring. We hope our responses below help clarify the remaining points.
>
> ---
> **Q1. Notation Refinement**
>
> Thank you for the valuable suggestion. We will revise the notation to use bold symbols for vectors (e.g., $\mathbf{x}$) and standard math notation for scalars (e.g., $x$) to improve clarity.
>
> ---
> **Q2. Difference Between Finetune and Seq-FT in Table 2**
>
> Thank you for the thoughtful question. The Finetune results refer to models trained independently on each task from scratch using LoRA fine-tuning, and evaluated only on that task. These results serve as upper bounds without any continual learning (CL) constraints. In contrast, Seq-FT is a vanilla CL baseline that performs sequential LoRA fine-tuning across the task sequence, without any specific mechanism to mitigate forgetting. As expected, it performs poorly on all three CL metrics.
>
> All CL methods (i.e., all rows except Zero-shot, Finetune, and Joint-learning) are evaluated under the same protocol: after training each new task, the model is tested on all tasks in our benchmark. This differs from traditional CL setups, which typically evaluate only on seen tasks. Since the pretrained MLLM has zero-shot capabilities, our protocol also assesses how well this ability is retained. The details of this evaluation protocol can be found in Appendix F. We will clarify this distinction in the revised version.
>
> ---
> **Q3. Training Efficiency Despite Architecture Evolution**
>
> Thank you for noticing this point. We also observe that D-MoLE brings an additional benefit in training efficiency, as shown in Table 6. While we briefly discussed this below the table, we would like to elaborate on it here.
>
> The main source of this efficiency gain lies in the selective placement of LoRA modules. Unlike methods such as joint-learning and O-LoRA, our method inserts LoRA modules only into a subset of the most sensitive transformer layers for each task. This design ensures that, although our method slightly increases the LoRA rank, the number of trainable parameters remains comparable to these baselines due to fewer insertion points.
>
> To further illustrate the efficiency, we conduct a toy experiment comparing GPU runtime under different LoRA ranks. Specifically, we generate a random input matrix of shape 64 × 1024 and perform 10,000 iterations of multiplying it with two simulated LoRA modules of rank 4 and rank 8, respectively. The measured average runtime per iteration is summarized below:
>
> **Table 1: GPU runtime comparison for different LoRA ranks.**
> |Rank|Avg. Runtime|
> |-|-|
> |r = 4|4.40 ms|
> |r = 8|4.70 ms (+6.8%)|
>
> Despite doubling the rank, the runtime increases by only 6.8%, suggesting that in low-rank settings, the startup overhead of matrix multiplication dominates the total runtime rather than the rank itself. Since our method inserts LoRA modules into fewer transformer layers, the total number of such operations is reduced. This compensates for the slightly higher cost per operation and results in a modest overall speedup.
>
> Moreover, our architecture evolution mechanism is lightweight by design. As shown in Table 7, the preprocessing time for computing the zero-cost proxy accounts for less than 2% of the total training time.
>
> We will make this explanation clearer in the revised version.
>
> ---
> **Q4. Load Balancing in Expert Routing**
>
> Thank you for the question. We do not explicitly introduce load balancing mechanisms in our routing design. Our routing mechanism is primarily intended for knowledge retention and transfer in the CL setting, rather than for inference efficiency or capacity scaling as in traditional MoE models.
>
> In our framework, expert collapse is unlikely to occur. During training, task IDs are known and each corresponding expert is explicitly activated and updated. During evaluation, task-level routing is based on reconstruction losses from task-specific autoencoders, each trained on its own task data. As shown in Appendix G, the reconstructed embeddings form well-separated clusters, enabling reliable expert selection and preventing collapse. The router operates on the entire multimodal instruction sequence, rather than on individual modalities. Therefore, varying modality distributions do not directly affect expert activation.
>
> In real-world deployments, especially under online CL settings, load balancing may become important. For example, some tasks may dominate the input stream, leading to expert overuse, while others remain underutilized. One possible solution is monitor expert activation frequency and incorporate regularization or usage-aware training objectives to maintain stable generalization. These strategies are orthogonal to our current design and may be worthwhile to explore in future work.

---

### Decision · Program_Chairs · 2025-05-01

**Decision:**

Accept (poster)

**Comment:**

This submission focuses on the multimodal instruction tuning under the context of continual learning. The authors proposed a D-MoLE method to explore the architecture evolving under the budgets by designing the LoRA Experts. Along with this, the authors introduced a gradient-based continual curriculum. Experiments show the 15% improvement on average over the best baseline.

Four reviewers commented this submission and finally their scores all reached to 4. Specifically, the authors substantially improved the experiments and made more clarification about some technical details and novelty. Two reviewers expressed their tendency for the rebuttal and the other two reviewers keep silence. After AC checked the rebuttal and the reviewers' questions, it seems that the authors rebuttal mostly considered the reviewers' questions. Hope the authors carefully incorporate the suggestions into the final revision.

Based on the reviewers' suggestion in ratings and the authors substantial rebuttal, AC tends to recommend acceptance about this submission.